# Ubiquitylation of nucleic acids by DELTEX ubiquitin E3 ligase DTX3L

Kang Zhu [ID] [1,2,6 ✉], Chatrin Chatrin [ID] [1,6], Marcin J Suskiewicz [ID] [3], Vincent Aucagne [ID] [3], Benjamin Foster [ID] [4], Benedikt M Kessler [ID] [5], Ian Gibbs-Seymour [ID] [4], Dragana Ahel [ID] [1 ✉] & Ivan Ahel [ID] [1 ✉]

## Abstract

**The recent discovery of non-proteinaceous ubiquitylation substrates broadened our understanding of this modification beyond conventional protein targets. However, the existence of additional types of substrates remains elusive. Here, we present evidence that nucleic acids can also be directly ubiquitylated via ester bond formation. DTX3L, a member of the DELTEX family E3 ubiquitin ligases, ubiquitylates DNA and RNA in vitro and that this activity is shared with DTX3, but not with the other DELTEX family members DTX1, DTX2 and DTX4. DTX3L shows preference for the 3′-terminal adenosine over other nucleotides. In addition, we demonstrate that ubiquitylation of nucleic acids is reversible by DUBs such as USP2, JOSD1 and SARS-CoV-2 PLpro. Overall, our study proposes reversible ubiquitylation of nucleic acids in vitro and discusses its potential functional implications.**

**Keywords** Ubiquitin; DTX3L; DELTEX; Nucleic Acid Modification; Genome Stability
**Subject Categories** Microbiology, Virology & Host Pathogen Interaction; Post-translational Modifications & Proteolysis; RNA Biology

## Introduction

Regulating protein function by adding ubiquitin (Ub), known as ubiquitylation or ubiquitination, is widely utilized by eukaryotic cells. Ubiquitylation is involved in nearly all aspects of cellular activities, ranging from protein degradation, which was the first function of ubiquitylation to be discovered, to immune signalling, DNA damage response, receptor trafficking and many more (Oh et al, 2018; Zheng and Shabek, 2017). Ubiquitylation is the sequential transfer of Ub to the ε-amino group of a lysine residue on the substrate by a Ub-activating enzyme (E1), a Ub-conjugating enzyme (E2) and a Ub ligase (E3), resulting in the formation of an iso-peptide bond between Ub C-terminal glycine and the acceptor lysine residue (Hershko and Ciechanover, 1998; Komander and Rape, 2012). In addition, some E3s catalyse Ub transfer to the hydroxyl groups of threonine and serine residues or a thiol group of a cysteine residue, forming oxyester or thioester bonds, respectively (Cadwell and Coscoy, 2005; Gao et al, 2021; Kelsall et al, 2019; Pao et al, 2018; Wang et al, 2009; Wang et al, 2007). Ub modification is highly reversible and detached by deubiquitinases (DUBs), with approximately a hundred of these enzymes encoded in the human genome (Mevissen and Komander, 2017).

Since ubiquitylation was first discovered five decades ago, tens of thousands of ubiquitylation sites on many proteins have been identified, indicating that most proteins are ubiquitylated spatio-temporally in cells. However, our understanding of ubiquitylation remains limited, particularly regarding the diverse types of ubiquitylation substrates. It has been assumed that the substrates of ubiquitylation are solely limited to proteins. However, several recent studies demonstrated that Ub can be covalently attached to non-proteinaceous substrates, such as lipopolysaccharide (LPS) (Otten et al, 2021), phosphatidylethanolamine (PE) (Sakamaki et al, 2022), or oligosaccharides (Kelsall et al, 2022). Furthermore, Ub can also be attached on another modification called ADP-ribosylation (Zhu et al, 2023; Zhu et al, 2022). ADP-ribosylation is a chemical modification of proteins and nucleic acids involving the addition of one or more ADP-ribosyl (ADPr) moieties. ADPr is transferred from nicotinamide adenine dinucleotide (NAD$^+$) onto the targets by ADP-ribosyltransferases (ARTs) including the best-studied PARPs (Groslambert et al, 2021; Pascal, 2018; Suskiewicz et al, 2023b). Hybrid ADPr-Ub modification—which consists of Ub attachment to ADPr and then as a whole conjugated to the substrates—is efficiently synthesized in vitro by the DELTEX family of E3s on both ADP-ribosylated proteins and nucleic acids substrates, allowing for indirect ubiquitylation (Kar et al, 2024; Zhu et al, 2023; Zhu et al, 2022), but its physiological relevance is not clear yet.

DELTEX-family E3 ligases have been suggested to be involved in many pathways, for example, Notch signalling, DNA damage repair, innate immune response and cancer progression, and have attracted significant attention over the last decade, but the exact mechanisms and physiological consequences have been elusive (Wang et al, 2021). DELTEX family in human is composed of five

[1] Sir William Dunn School of Pathology, University of Oxford, Oxford, UK. [2] Health Science Center, East China Normal University, Shanghai 200241, China. [3] Centre de Biophysique Moléculaire, CNRS UPR 4301, Orléans, France. [4] Department of Biochemistry, University of Oxford, South Parks Road, Oxford, UK. [5] Target Discovery Institute, Centre for Medicines Discovery, Nuffield Department of Medicine, University of Oxford, Oxford OX3 7FZ, UK. [6] These authors contributed equally: Kang Zhu, Chatrin Chatrin.
✉E-mail: kang.zhu@path.ox.ac.uk; dragana.ahel@path.ox.ac.uk; ivan.ahel@path.ox.ac.uk

members, namely DTX1, DTX2, DTX3, DTX4 and DTX3L (Takeyama et al, 2003; Wang et al, 2021). Different family members have distinct N-terminal domains, either WWE domains (in DTX1, DTX2 and DTX4), or KH domain(s) (in DTX3 and DTX3L) (Fig. 4A) (Zhu et al, 2023). WWE domains in proteins are known to bind poly(ADP-ribose) chains (DaRosa et al, 2015) whereas KH domains are known for binding to single-stranded nucleic acids (Nicastro et al, 2015; Suskiewicz et al, 2023a; Valverde et al, 2008). In addition, DTX3L contains an RNA recognition motif (RRM) domain preceding the KH domains. In contrast to the varied N termini, DELTEX E3s share a characteristic C-terminal tandem RING-DTC (RD) domains, where the RING domain acts as an E3 Ub ligase and the DTC domain has been demonstrated to bind $NAD^+$ and ADPr through its conserved pocket (Chatrin et al, 2020). Like other RING-type E3s, DELTEX RING domains do not determine the specificity of Ub acceptors (amino or hydroxyl group), as canonical RING-type E3s leave the task of defining acceptor specificity to the E2 enzyme that they recruit (Wenzel et al, 2011).

DELTEX E3s evolved to have an accompanying DTC domain adjacent to RING domain, which bind $NAD^+$ or ADPr and provide two catalytic residues to enable $NAD^+$ or ADPr ubiquitylation on their 3′ hydroxyl groups of the adenine-proximal ribose (Fig. 3 in (Zhu et al, 2022)). Mechanistically, DELTEX E3s recruit E2~Ub conjugate and one $NAD^+$ or ADPr molecule using the RING and the DTC domains, respectively. Next, the thioester bond between E2 and Ub is juxtaposed to the 3′ hydroxyl group of $NAD^+$ or ADPr proximal ribose due to the flexible linker between the RING domain and the DTC domain (Chatrin et al, 2020; Zhu et al, 2022). Because the hydroxyl moiety is a weak nucleophile, the DTC domain contributes one histidine residue and one glutamate residue to apparently deprotonate and thus encourage the 3′ hydroxyl group to attack the E2~Ub conjugate to accomplish ADPr ubiquitylation.

Given the similarity between ADPr and nucleic acids, which are composed of the same constituents: nucleobases (adenine), ribose sugars and phosphates, we became intrigued by the possibility of nucleic acids becoming directly ubiquitylated. Indeed, in this study, we demonstrate that DTX3L, representing the KH domain-containing DELTEX E3s, ubiquitylates nucleic acids in vitro. The modification occurs primarily on the 3′-terminal adenosine nucleotides, likely targeting the 3′ hydroxyl group of the ribose sugar, resulting in an ester bond formation. In vitro, ubiquitylation of nucleic acids on the 3′ adenosine nucleotide protects them from degradation by 3′ → 5′ nucleases. Lastly, we show—also in vitro— the reversibility of the DTX3L-mediated nucleic acids ubiquitylation by some DUBs including USP2, JOSD1 and SARS-CoV-2 PLpro. Our study establishes nucleic acids as a novel type of ubiquitylation substrate.

## Results and discussion

### DTX3L RING-DTC ubiquitylates nucleic acids carrying the 3′ adenosine nucleotide

In our previous studies, we showed that the bipartite RD domains of DELTEX family E3s are capable of ubiquitylating ADPr on the 3′ hydroxyl group of the adenine-proximal ribose (Zhu et al, 2023; Zhu et al, 2022). Specifically, the DTC domain first accommodates

the ADPr molecule to position the 3′ hydroxyl group of ADPr proximal ribose close to the E2~Ub conjugate bound by the RING domain. The DTC domain appears to then utilize its two crucial catalytic residues to deprotonate the 3′ hydroxyl group, thus facilitating ADPr ubiquitylation. The available experimental structures show that the DTC domain uses the same conserved pocket to bind either ADPr or $NAD^+$, and can facilitate Ub transfer to both (Fig. 1A) (Ahmed et al, 2020; Chatrin et al, 2020; Zhu et al, 2023). Both ADPr and $NAD^+$ contain an AMP moiety (Appendix Fig. S1), and it is this part that becomes ubiquitylated on the 3′ hydroxyl. By analysing the ADPr/DTX2-RD and the $NAD^+$/DTX1-RD structures, we found in both complexes the shared AMP part of ADPr and $NAD^+$ inserted into a deep pocket of the DTC domains (Fig. 1A). The AMP moiety has a highly similar conformation in the two structures and makes close contacts with the neighbouring amino residues, unlike the distal ribose of ADPr/$NAD^+$, which protrudes out of the binding pocket, showing fewer contacts with the DTC domain. This finding prompted us to test whether AMP itself is also a substrate for ubiquitylation by DELTEX E3s. To test this, we utilized high-performance liquid chromatography coupled to mass spectrometry (HPLC-MS) to analyse the ubiquitylation reaction of DTX3L-RD with AMP, which shows that Ub is 100% converted into Ub-AMP (Appendix Fig. S2). This suggests that AMP is a substrate for ubiquitylation by DELTEX E3s, and its ubiquitylation efficiency is comparable to that of ADPr or $NAD^+$, where more than 90% of the starting Ub is conjugated to ADPr or $NAD^+$ under the same conditions (Appendix Fig. S3) (Zhu et al, 2022).

Considering that AMP or 2′ deoxy-AMP (dAMP) are building blocks for RNA or DNA, nucleic acids ending with adenosine nucleotide at the 3′ end will present an AMP/dAMP moiety with a free ribose 3′ hydroxyl, thus representing a potential ubiquitylation substrate for DELTEX E3s (Fig. 1B). We wondered if RNA/DNA ending with (deoxy-)adenosine, could become directly ubiquitylated on their terminal riboses. To test this possibility, we designed a Cy3-labelled 21-nucleotide-long single-stranded DNA (ssDNA) with 3′ deoxy-adenosine nucleotide (E21_DNA_A). Again, we used DTX3L (its RD fragment) as an E3 ligase, because the presence of multiple single-stranded nucleic acids-binding domains in the full-length form of this DELTEX-family member suggests a functional link with nucleic acids (Zhu et al, 2023). We incubated DTX3L-RD with E21_DNA_A and Ub-processing components (E1, E2, Ub and ATP) and resolved the reaction mixtures on both SDS-PAGE and 20% TBE-Urea gels to visualize the potential nucleic acids-Ub adducts. As expected, in the presence of all ubiquitylation components, an upward-shifted band appeared, indicating that E21_DNA_A became ubiquitylated (Fig. 1C, lane 2). However, the reactions omitting any ubiquitylation component did not show any higher band (Fig. 1C, lane 3–6), which is consistent with what was observed for ADPr ubiquitylation (Zhu et al, 2023; Zhu et al, 2022). Similarly, we then used an ssRNA substrate that has the same sequence as E21_DNA_A (E21_RNA_A), and showed that E21_RNA_A was also ubiquitylated by DTX3L-RD (Fig. 1D, lane 2). Full-length DTX3L (DTX3L fl) appears to be more efficient than DTX3L-RD in catalysing E21_DNA_A ubiquitylation, possibly owing to enhanced substrate recruitment through the mentioned multiple nucleic acids-binding domain (Zhu et al, 2023) (Fig. EV1). However, since the minimum catalytic RING-DTC (RD) fragment is proficient enough and easier to produce, this fragment is used

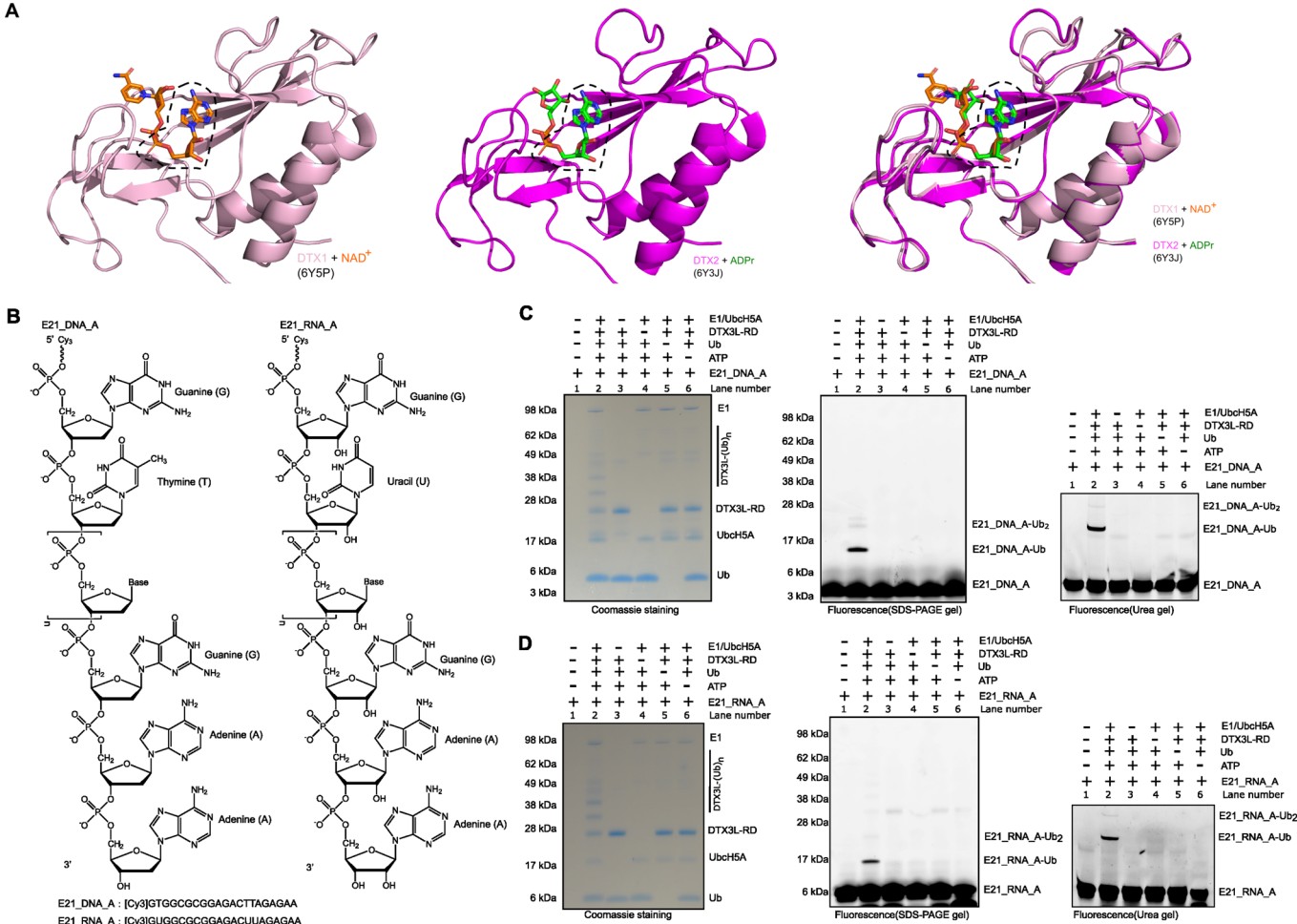

**Figure 1. Biochemical characterization of DTX3L-RD-catalysed nucleic acids ubiquitylation.**

(A) Structural analysis showing that the shared AMP moiety in NAD[+] and ADPr inserts deeply in the binding pockets of the DTC domains from DTX1 or DTX2. The AMP moiety is indicated by black dashed box. (B) Chemical structures of the E21_DNA_A and E21_RNA_A used in this study. (C) Biochemical reconstitution of E21_DNA_A ubiquitylation. E21_DNA_A-Ub was obtained by incubation of DTX3L-RD and E1, E2 UbcH5A, ATP and Ub. Omitting any of these components blocked the E21_DNA_A ubiquitylation. The reactions were divided into two parts. One part was analysed on an SDS-PAGE gel and visualized by first Molecular Imager PharosFX system (BioRad) and then Coomassie staining. Another part was loaded on a pre-run 20% denaturing urea PAGE gel. The gels were run at 6 W/gel and following visualization using the Molecular Imager PharosFX system (BioRad). (D) As in (B), E21_RNA_A was used as substrate for the ubiquitylation reactions. Each experiment has been completed in triplicate.

throughout the study. We further tested whether DTX3L-RD could ubiquitylate double-stranded (ds) nucleic acids. Unlike single-stranded nucleic acids, which were ubiquitylated, neither dsDNA nor dsRNA were ubiquitylated by DTX3L-RD (Fig. EV2). This indicates that DTX3L-RD specifically ubiquitylates single-stranded nucleic acids ending with 3′-adenosine nucleotides. This agrees with the fact that, based on the structure of the ADPr-bound DTX2-RD, the adenine base is tightly encapsulated within the DTC domain in a way that should be sterically incompatible with base-pairing in the context of a double-stranded nucleic acids (Fig. 1A).

Next, we wanted to figure out on which chemical moiety within nucleic acids Ub is attached. Considering the chemistry of the ubiquitylation reaction, with E2~Ub acting as an electrophile, we focused on nucleophilic moieties that could act as Ub acceptors. Depending on whether DNA or RNA is used, one or two hydroxyl group(s) on the 3′-terminal adenosine are available, in addition to

several amine groups on terminal or internal bases (Fig. 1B). However, considering the similarity between adenosine nucleotide and ADPr, the 3′ hydroxyl group of the 3′-terminal ribose appears the most likely candidate (Zhu et al, 2023). Of note, the 3′ hydroxyl group, unlike the 2′ hydroxyl group, is shared between DNA and RNA molecules, both of which were efficiently ubiquitylated above. We used phosphorylation to block the 3′ hydroxyl group of the terminal ribose, and tested if it affected the Ub modification. We conducted ubiquitylation reactions using DTX3L-RD and E21_DNA_A as well as its 3′ phosphorylated form, E21_DNA_A_3P. In contrast to E21_DNA_A, which was ubiquitylated, E21_DNA_A_3P ubiquitylation was greatly weakened, suggesting that the terminal 3′ hydroxyl group is the likely Ub acceptor site (Fig. 2A). The weak remaining ubiquitylation of the phosphorylated DNA might be—at least in part—due to incomplete phosphorylation. Since synthesizing the nucleic acids without

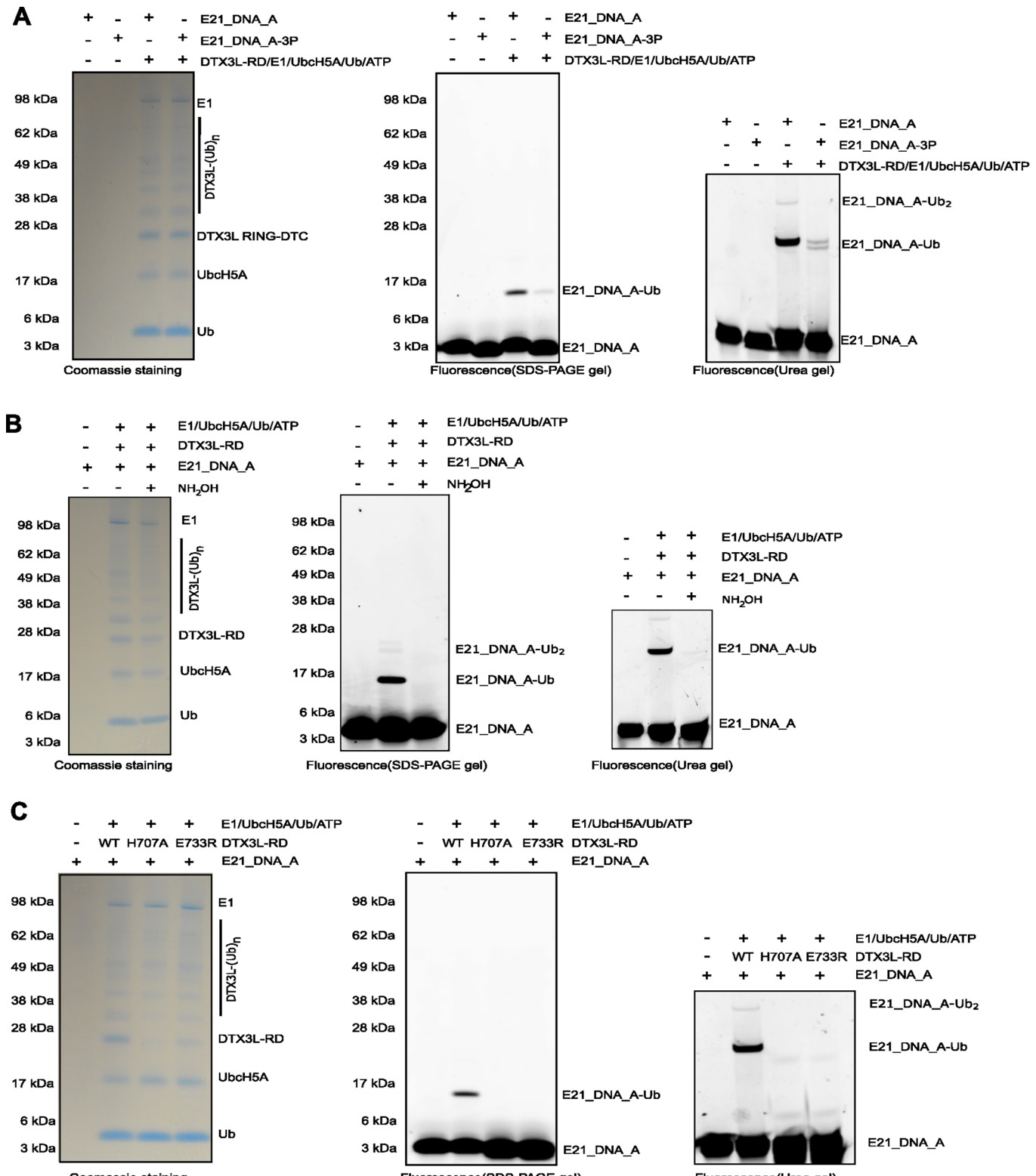

**Figure 2. DTX3L-RD attaches Ub onto the 3′ hydroxyl group of terminal adenosine in nucleic acids.**

(A) 3′ phosphorylation markedly reduced the ubiquitylation of DNA by DTX3L-RD. E21_DNA_A and its 3′ phosphorylation form: E21_DNA_A-3P were incubated with E1, E2 UbcH5A, ATP and Ub. The reactions were analysed on an SDS-PAGE gel and urea PAGE gel and processed as described before. (B) $NH_2OH$ reverses DTX3L-RD-catalysed E21_DNA_A ubiquitylation. $NH_2OH$ cleaves the ester bond between the carbonyl group of $Gly^{76}$ of Ub and the 3′ hydroxyl group of the A of nucleic acids. (C) DTX3L-RD ADPr ubiquitylation inactive mutants failed to produce upshift bands that correspond to ubiquitylation of DNA, indicating that Ub is attached to 3′ hydroxyl group. Each experiment has been completed in triplicate.

the 3′ hydroxyl group is technically challenging, we used 2′-deoxy ATP and 3′-deoxy ATP as model substrates for verifying the Ub acceptor specificity of DTX3L. Indeed, HPLC-MS analysis showed that DTX3L-RD significantly ubiquitylated 2′-deoxy ATP. On the contrary, no trace of 3′-deoxy ATP ubiquitylation was detected, again suggesting that DTX3L-RD targets the 3′ hydroxyl group of 3′ terminal ribose of E21_DNA_A and E21_RNA_A (Appendix Figs. S4 and S5). Considering that the ester bond between the Gly$^{76}$ residue of Ub and the 3′ hydroxyl group of terminal adenosine nucleotide should be sensitive to NH$_2$OH treatment (Zhu et al, 2023; Zhu et al, 2022), we used NH$_2$OH to see if it can reverse the ubiquitylation of E21_DNA_A and E21_RNA_A. Our result showed that NH$_2$OH completely removed the ubiquitylation, speaking against the possibility of amide group-linked ubiquitylation, which is resistant to NH$_2$OH (Fig. 2B, Fig. EV3A). Consistent with this, ubiquitylation of E21_DNA_A and E21_RNA_A was also abolished upon mutating catalytic histidine and glutamate residues (H707A and E733R) present in the DTC domain of DTX3L-RD (Figs. 2C and EV3B), which are required for ADPr ubiquitylation on 3′ hydroxyl but not canonical lysine ubiquitylation (Zhu et al, 2023; Zhu et al, 2022).

Overall, these results suggest that the observed ubiquitylation of nucleic acids happens on the 3′-terminal adenosine nucleotide, likely through its 3′ hydroxyl group.

## DTX3L-RD shows preference for 3′-terminal adenosine nucleotide over other nucleotides in nucleic acids

Since we showed that DTX3L-RD could ubiquitylate the 3′ adenosine (A) of nucleic acids, we wondered whether other nucleotides at the 3′ end could also be ubiquitylated. According to their chemical structures, the purine GMP resembles AMP with a two-ring structure, while pyrimidines CMP and TMP have a different one-ring structure (Fig. 3A). To test our idea, we first utilized HPLC-MS to analyse the ubiquitylation reaction of DTX3L-RD with free nucleotides including GMP, CMP and TMP (Appendix Figs. S6–8), using AMP and ADPr as the control (Appendix Figs. S2–3). Interestingly, we detected the masses that are consistent with the molecular weights of Ub-GMP, Ub-CMP and Ub-TMP, but the efficiency of ubiquitylation, judged by the percentage of nucleotides that became modified, varied and was generally lower than for Ub-AMP which was the only nucleotide that can be quantitatively modified (Fig. 3B) indicating that AMP is preferred by DTX3L-RD. In all cases, the only Ub-containing products detected were the starting Ub and Ub-NMP (N = A, T, C or G), together with an Ub-DTT adduct in the case of the poor substrates GMP, TMP, and CMP. This adduct is likely formed upon a direct nucleophilic attack of the E2~Ub conjugate by DTT, through a trans-thioesterification process.

Next, we wanted to know whether DTX3L-RD could ubiquitylate nucleic acids with 3′ G, T or C ends and whether the efficiency of the potential ubiquitylation shows a similar trend to that observed with free nucleotide monophosphates. We redesigned the E21_DNA_A to have different 3′ ends, namely, T, C and G ends, and tested these in our ubiquitylation assay. Our results showed the most abundant ubiquitylation for E21_DNA_A, followed by E21_DNA_G, while only a small fraction of E21_DNA_T and E21_DNA_C were ubiquitylated (Fig. 3C). This data is consistent with our MS data obtained with free nucleotides.

Taken together, our results suggest that DTX3L preferably ubiquitylates 3′-terminal adenosine of nucleic acids over other 3′-terminal nucleotides. In addition, the strict dependence of the reaction efficiency on the nature of the 3′-terminal nucleotide further supports the notion that the modification takes place on that nucleotide.

## DTX3 and DTX3L possess nucleic acids ubiquitylation activity whilst DTX1, DTX2 as well as DTX4 do not

Based on their N-terminal domains, DELTEX family E3s can be divided into two sub-classes (Fig. 4A): (1) WWE domain-containing DELTEX E3s: DTX1, DTX2 and DTX4, all of which possess two WWE domains (Wang et al, 2021; Zweifel et al, 2005); and (2) KH domain-containing DELTEX E3s: DTX3 and DTX3L, where DTX3 contains one KH domain while DTX3L has five KH domains (Zhu et al, 2023). Considering that the WWE domain typically binds the poly(ADP-ribose) chain (DaRosa et al, 2015; Kang et al, 2011; Wang et al, 2012), while the KH domain tends to bind single-stranded nucleic acids (Nicastro et al, 2015; Valverde et al, 2008), we speculated that this might reflect a functional difference, with KH domain-containing DELTEXes having a nucleic acids-related function. Our results so far indicated that DTX3L-RD ubiquitylates nucleic acids, preferably on its 3′-terminal adenosine. We wondered whether other DELTEX family members are capable of ubiquitylating nucleic acids. Initially, we tested the DTX2, a WWE domain-containing DELTEX, observing that DTX2-RD did not show any ubiquitylation activity on E21_DNA_A, even at 16 µM enzyme concentration, whilst DTX3L-RD ubiquitylated E21_DNA_A at 1 and 4 µM concentration (Fig. EV4). Since, in the above experiments, the RD fragments of DELTEX E3s were used, the ability to modify nucleic acid does not come from the presence or absence of KH domains, but rather is inherent to the RD fragment. Interestingly, when using the poly (ADP-ribose) chain (PAR) as a substrate, we did not observe any difference in the activities of DTX3L-RD and DTX2-RD (Fig. 4B,C).

Next, we analysed the DNA ubiquitylation by the RD fragments of all human DELTEX family members and revealed that DTX3-RD and DTX3L-RD ubiquitylated E21_DNA_A whilst DTX1-RD, DTX2-RD and DTX4-RD did not (Fig. 4D). This suggests that KH domain-containing DELTEX E3s: DTX3 and DTX3L exhibit nucleic acids ubiquitylation activity, whereas WWE domain-containing DELTEX E3s: DTX1, DTX2 and DTX4 lack this capability. Taken all together, the RD fragments in DELTEX E3s determine the nucleic acids ubiquitylation ability and apparently evolved differently in the two sub-classes of DELTEX E3 ligases, which correlates well with the presence of N-terminus KH domains. It is possible that the KH domains in full-length DTX3 and DTX3L can further provide specificity and efficiency for nucleic acids modification.

Considering the differences between RD fragments of KH domain-containing DELTEXes (DTX3 and DTX3L) and those of WWE domain-containing DELTEXes (DTX1, DTX2 and DTX4), we speculated that the slightly shorter RING domain and/or the lack of an alanine-arginine (AR) insertion in the DTC domains of DTX3 and DTX3L might affect their ability in ubiquitylating nucleic acids (Appendix Fig. S9A–C). We inserted AR residues to DTX3L-RD (DTX3L-RD$^{ins\_AR}$), deleted the AR of DTX2-RD (DTX2-RD$^{del\_AR}$) and performed the ubiquitylation reactions. Fittingly, DTX3L-RD$^{ins\_AR}$ lost the nucleic acids ubiquitylation activity. However, DTX2-RD$^{del\_AR}$ did not acquire the capability for nucleic acids ubiquitylation (Appendix

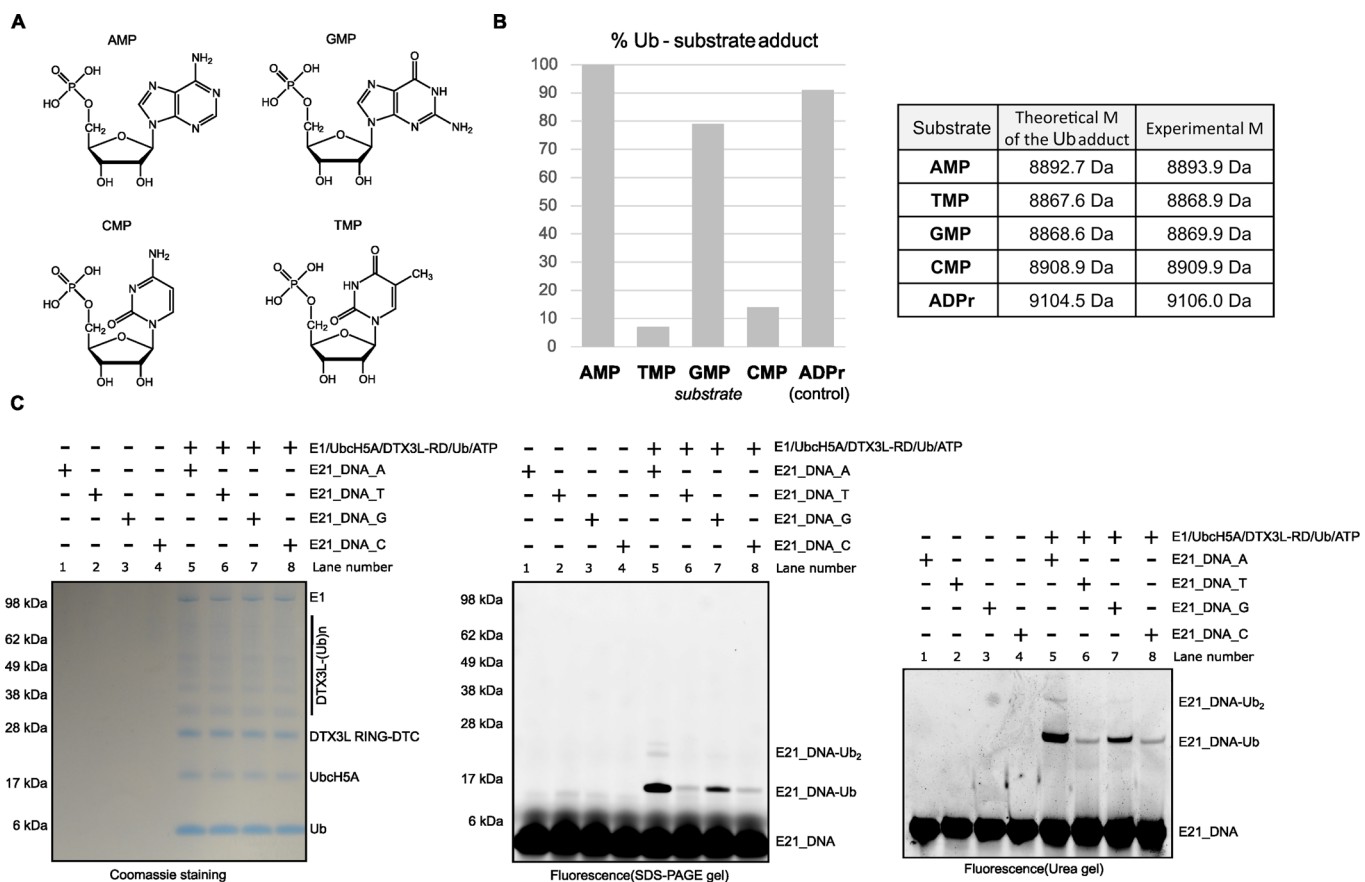

**Figure 3. DTX3L-RD preferably ubiquitylates 3′ adenosine base of nucleic acids.**

(A) Chemical structures of different nucleotides. (B) HPLC-MS-based identification of the products of DTX3L-RD-catalysed ubiquitylation reactions performed with indicated nucleotides. Detected average masses and theoretical ones are provided. Ub was used at a 25 µM concentration, and all substrates at 12 mM (48 molar equivalents). $n = 1$. (C) DTX3L-RD ubiquitylates 3′ A, G, T and C of nucleic acids. E21_DNA_A-Ub was obtained by incubation of DTX3L-RD and E1, E2 UbcH5A, ATP and Ub. The reactions were divided into two parts. One part was analysed on an SDS-PAGE gel and visualized by first florescence and then Coomassie staining. Another part was loaded on a pre-run 20% denaturing urea PAGE gel. The gels were run at 6 W/gel and following visualization using the Molecular Imager PharosFX system (BioRad). This experiment has been completed in triplicate.

Fig. S9D). These mutants were both active in autoubiquitylation but were impaired in ADPr ubiquitylation (Appendix Fig. S9E). These results suggest that the absence or presence of AR residues in DTX3L and DTX2 are crucial for both nucleic acids and ADPr ubiquitylation. In NAD$^+$ or ADPr ubiquitylation, the catalysis is strictly coordinated by the RING domain, the DTC domain, and their specific orientation via the flexible linker between them (Chatrin et al, 2020). It is likely that nucleic acids ubiquitylation also follows a similar principle, with specific structural requirements within and possibly between the two domains. It is likely that other unidentified elements may also be essential. Hence, a more detailed investigation is warranted for a comprehensive understanding of the difference between the two DELTEX sub-classes with respect to their ubiquitylation substrate specificity.

## Ubiquitylation of nucleic acids prevents degradation and is reversible

Chemical modifications on nucleic acids play important roles in their function including influencing their stability. For example, eukaryotic mRNAs undergo co-transcriptional modification through the addition of a 7-methylguanosine cap (m7G), which shields mature mRNAs from degradation by $5′ \rightarrow 3′$ exonucleases (Furuichi et al, 1977; Shatkin, 1976). In contrast, NAD$^+$ capping at the 5′ end of RNA has been observed to promote degradation (Jiao et al, 2017; Yu et al, 2021). Recent studies have reported ADPr as another capping mechanism of the 5′ end of RNA, which protects them from degradation by nucleases, thus improving their stability (Munnur et al, 2019). Prompted by these findings, we investigated whether the ubiquitylation of nucleic acids at their 3′ end influences their stability. We first used DTX3L-RD and E21_RNA_A to generate ubiquitylated E21_RNA_A, then treated the reaction mix with $3′ \rightarrow 5′$ exonuclease T (Exo T). We observed that unmodified E21_RNA_A was completely degraded, while the ubiquitylated E21_RNA_A was resistant to Exo T treatment (Fig. 5A), suggesting the protective role of ubiquitylation in this specific manner.

Following the characterization of nucleic acids ubiquitylation by DTX3L, we next wanted to test whether the Ub modification

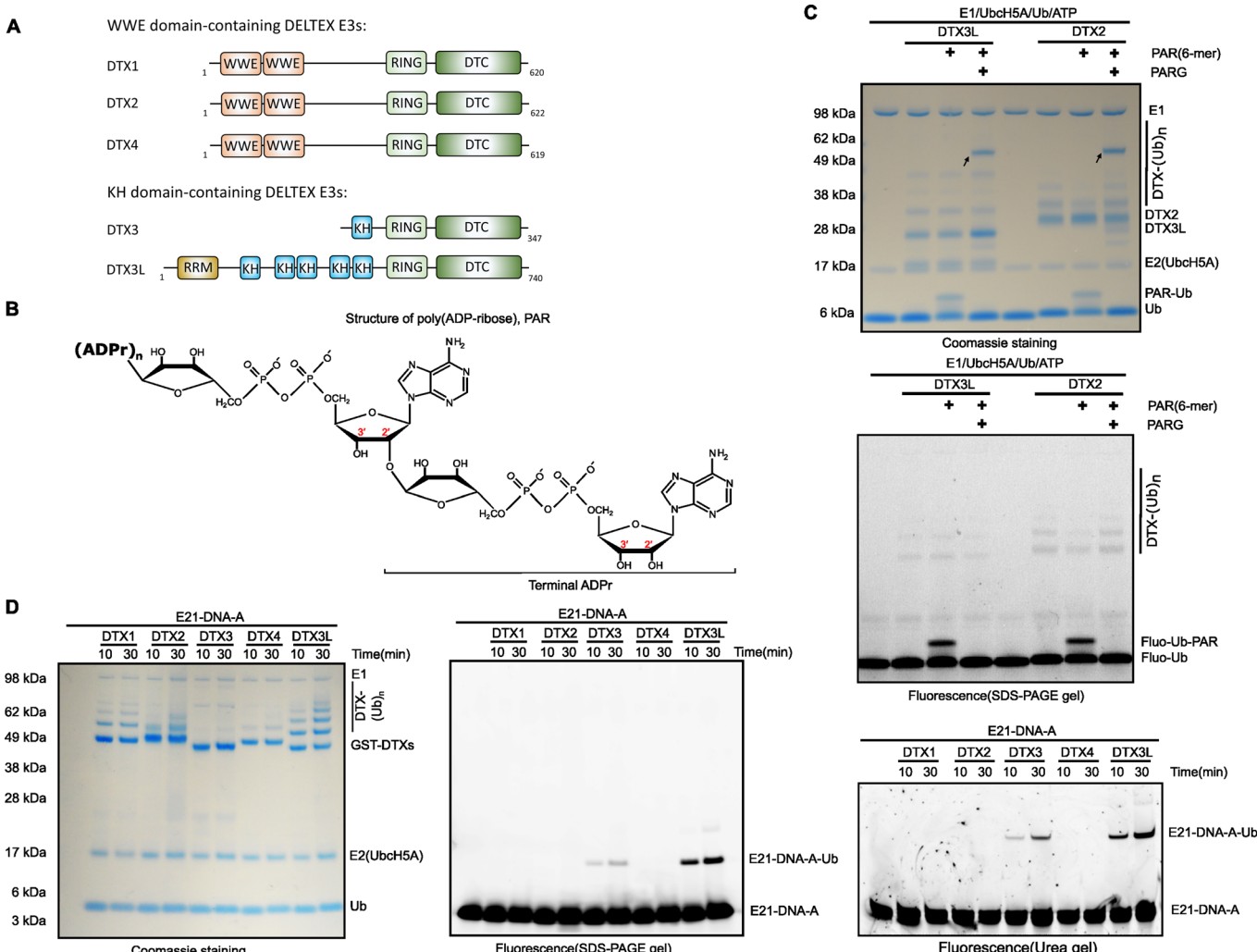

**Figure 4. Differences between WWE domain-containing and KH domain-containing DELTEX E3s.**

(A) Domain organisation of DELTEX family E3s. DTX1, DTX2 and DTX4 are classified into WWE domain-containing DELTEXes; DTX3 and DTX3L are classified into KH domain-containing DELTEXes. (B) Chemical structure of poly(ADP-ribose) chain, PAR. (C) DTX2-RD and DTX3L-RD ubiquitylate PAR. The arrows indicate PARG. (D) DTX3-RD and DTX3L-RD, but not DTX1-RD, DTX2-RD as well as DTX3L-RD, are able to ubiquitylate nucleic acids. E21_DNA_A was incubated with E1, E2 UbcH5A, ATP, Ub and individual DELTEX E3s' RD domain, then the reactions were analysed on an SDS-PAGE gel and Urea gel and visualized using the Molecular Imager PharosFX system (BioRad). Each experiment has been completed in triplicate.

on nucleic acids is a reversible process. Modifications of proteins and nucleic acids can typically be reversed by the action of so-called eraser enzymes (DUBs in the case of ubiquitylation), which can remove the added chemical groups. We first generated ubiquitylated E21_DNA_A using DTX3L-RD and then treated the reaction mix with USP2, an Ub-substrates linkage-nonspecific DUB (Amerik and Hochstrasser, 2004; Renatus et al, 2006). Notably, USP2 but not its catalytically inactive mutant, removed the Ub modification from E21_DNA_A (Fig. 5B). In addition, we also included SARS2-CoV-2 PLpro, since SARS2 virus infection causes a strong induction of DTX3L, which has been reported to function as an antiviral protein (Heer et al, 2020). Our results showed that SARS-CoV-2 PLpro reversed ubiquitylation of E21_DNA_A completely, but its catalytically inactive mutant did not (Fig. 5B). In an analogous set of experiments, we reproduced the same observations with the ssRNA E21_RNA_A

(Fig. 5C). In addition, we extended these experiments to more DUBs, including OTULIN, OTUB1, and OTUD2, which exhibit isopeptidase activity, and JOSD1, which possesses esterase activity (De Cesare et al, 2021). We found that JOSD1 reversed DNA ubiquitylation, which is an ester linkage between DNA's 3′ terminal adenosine base and Ub C-terminus. In contrast, OTULIN, OTUB1, and OTUD2, failed to cleave the ubiquitylated DNA (Fig. EV5). This data suggests that DTX3L-mediated nucleic acid ubiquitylation is reversed by DUBs with esterase activity.

Taken together, our data show that ubiquitylation of nucleic acids on the 3′ adenosine can function as a protective mechanism against 3′ → 5′ nucleases such as ExoT in vitro, suggesting the potential protective role in cells. Furthermore, we demonstrated that nucleic acids ubiquitylation can be enzymatically reversed by USP2, JOSD1 and SARS-CoV-2 PLpro.

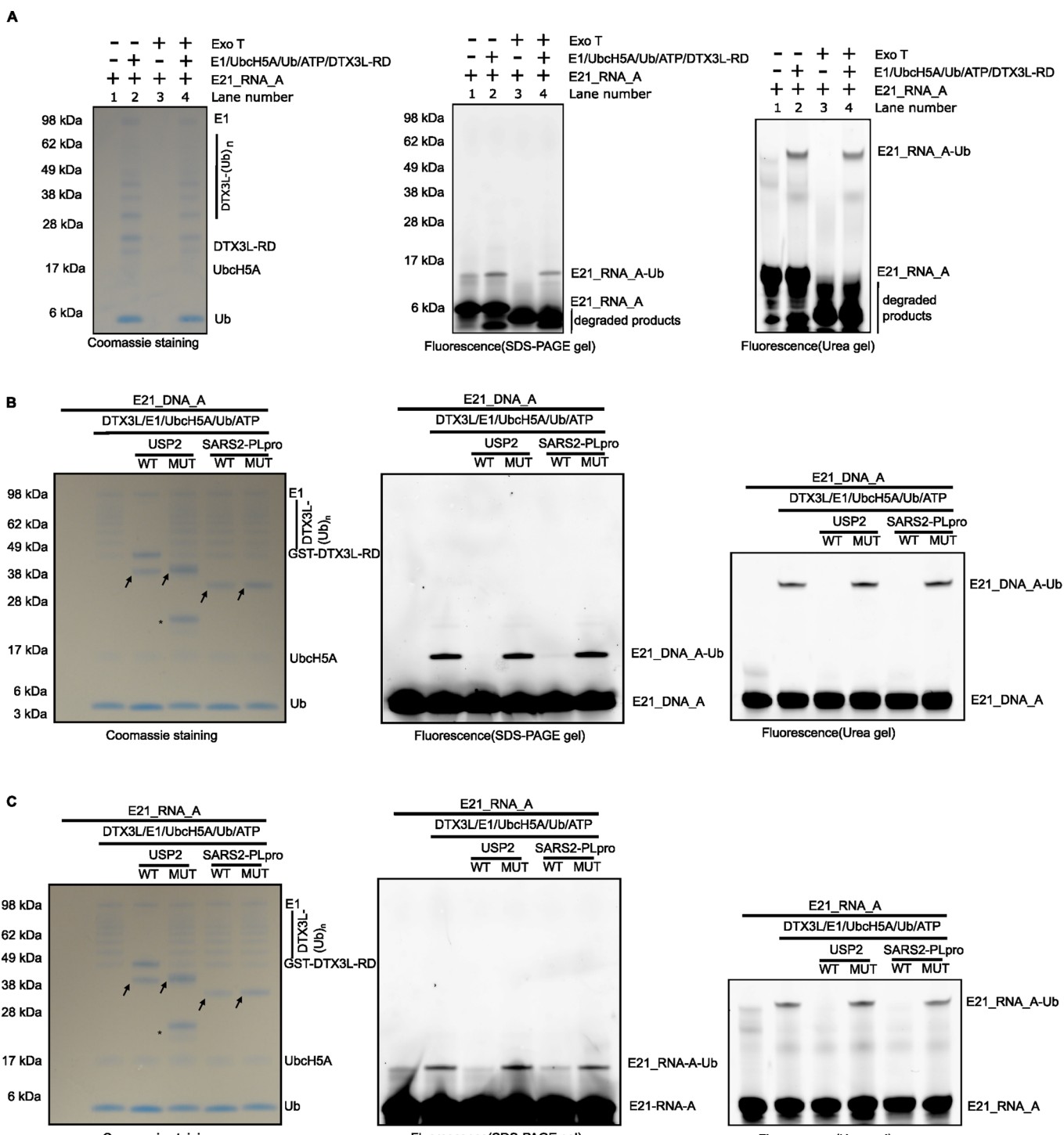

**Figure 5. Ubiquitylation of nucleic acids protect them from 3′ → 5′ nuclease attack and is a reversible process.**

(A) The ubiquitylation of E21_RNA_A prevents cleavage by the 3′-5′ nuclease Exo T. Ubiquitylated E21_RNA_A and unmodified E21_RNA_A were treated with Exo T and the reactions were resolved via SDS-PAGE gel and Urea gel. RNA and RNA-Ub adducts were visualised by the Molecular Imager PharosFX system (BioRad). (B) Hydrolysis of ubiquitylated nucleic acids. Following the ubiquitylation of E21_DNA_A with DTX3L-RD, the indicated wild type and catalytically inactive mutant DUBs were added and further incubated. The reactions were analysed and visualized as described earlier. The arrows indicate various hydrolases. The asterisks indicate contamination in USP2 catalytically inactive mutant. (C) As in (B), ubiquitylated E21_RNA_A was used as substrates for hydrolysis. The reactions were analysed and visualized as described earlier. Each experiment has been completed in triplicate.

## Conclusion

Although nucleic acids modifications have traditionally been associated with small chemical groups, such as methyl (for example, in the case of N6-methyladenosine, m6A), our investigation demonstrates that nucleic acids can—at least in vitro—undergo efficient and reversible modification with a large, proteinaceous modifier (Ub) by the action of certain E3 ligases and DUBs. Proving the existence of nucleic acids ubiquitylation in cells by either DTX3 and DTX3L or by some of the thousands of other uncharacterized ubiquitylation enzymes across various organisms warrants further investigation. We can speculate at this point that ubiquitylation of nucleic acids might have various in vivo implications: the protection of nucleic acids' 3′ ends, blocking DNA end processing during DNA repair, influencing nucleic acids' stability, interactome, and translational processes. However, it is also possible that nucleic acids ubiquitylation is a detrimental off-target activity of DTX3 and DTX3L, which—when it occurs in the cell—would need to be repaired. In such a scenario, DUBs such as USP2 would function as repair enzymes by reversing the aberrant ubiquitylation of nucleic acids, analogously to the repair of DNA adenylates formed during abortive DNA ligation events by aprataxin (APTX) (Ahel et al, 2006). Similarly, it has been suggested that several PARPs mistakenly ADP-ribosylate DNA, thereby generating DNA lesions (adducts), which could be repaired by ADPr hydrolases (Munnur and Ahel, 2017). Future studies should clarify these aspects. Better tools and analytical techniques, such as specific antibodies or advanced mass spectrometry approaches, will need to be developed to support these future efforts in understanding the functional significance of nucleic acids ubiquitylation.

## Methods

### Reagents and tools table

| Reagent/Resource | Reference or Source | Identifier or Catalog number |
|---|---|---|
| **Chemicals, enzymes and other reagents** | | |
| Benzonase | Novagen | |
| cOmplete EDTA-free protease inhibitor cocktail | Roche | |
| Superdex 200 10/300 | GE Healthcare | |
| Ni$^{2+}$-NTA agarose resin | Serva | |
| UBE1 | R&D Systems | E-304-050 |
| UbcH5A | R&D Systems | E2-616-100 |
| Ub | R&D Systems | U-100H-10M |
| ExoT nuclease | NEB | M0265S |
| DTX3L fl | Zhu et al, 2023 | |
| SARS-CoV-2 PLPro | Zhu et al, 2023 | |
| USP2 | Zhu et al, 2023 | |

| Reagent/Resource | Reference or Source | Identifier or Catalog number |
|---|---|---|
| His-DTX2-RD | Zhu et al, 2022 | |
| His-DTX3L-RD | Zhu et al, 2022 | |
| Fluorescein-Ub | Cambridge Bioscience | 7552-50 |
| ATP | Thermo Fisher | R0441 |
| NAD$^+$ | Merck | 10127965001 |
| Biotin-NAD$^+$ | Bio-Techne | 6573 |
| ADP-ribose | Sigma | A0752 |
| AMP | Santa Cruz Biotechnology | sc-214504A |
| TMP | Sigma | T7004 |
| GMP | Abcam | ab146534 |
| CMP | Sigma | C1006 |
| 2′-deoxy ATP | MP Biomedicals | 0210054925 |
| 3′-deoxy ATP | Santa Cruz Biotechnology | sc-396713 |
| **Recombinant DNA** | | |
| pDEST15-DTX1(388–620) | This study | |
| pDEST15-DTX2(390–622) | This study | |
| pDEST15-DTX3(148–347) | This study | |
| pDEST15-DTX4(387–619) | This study | |
| pDEST15-DTX3L(544–740) | This study | |
| pDEST17-JOSD1 | This study | |
| **Antibodies** | | |
| Ubiquitin Antibody (P4D1) mouse monoclonal | Santa Cruz Biotechnology | sc-8017 |
| IRDye® 800CW goat anti-mouse | Li-Cor | 926-32210 |
| NeutrAvidin DyLight™ 680 | Thermo Fisher | Cat # 22848 |
| **Oligonucleotides and other sequence-based reagents** | | |
| 5Cy3_E21_DNA_A [Cy3] GTGGCGCGGAGACTTAGAGAA (PAGE) | This study, Sigma-Aldrich | |
| 5Cy3_E21_DNA_T [Cy3] GTGGCGCGGAGACTTAGAGAT (PAGE) | This study, Sigma-Aldrich | |
| 5Cy3_E21_DNA_G [Cy3] GTGGCGCGGAGACTTAGAGAG (PAGE) | This study, Sigma-Aldrich | |
| 5Cy3_E21_DNA_C [Cy3] GTGGCGCGGAGACTTAGAGAC (PAGE) | This study, Sigma-Aldrich | |
| 5Cy3_E21_DNA_A_3′P [Cy3] GTGGCGCGGAGACTTAGAGAA[Phos] (PAGE) | This study, Sigma-Aldrich | |
| 5Cy3_E21_DNA_A_3′P [Cy3] GTGGCGCGGAGACTTAGAGAA[Phos] (HPLC) | This study, Sigma-Aldrich | |
| 5Cy3_E21_RNA_A [Cy3] GUGGCGCGGAGACUUAGAGAA (PAGE) | This study, Sigma-Aldrich | |
| **Other** | | |
| EmulsiFlex-C5 homogenizer | Avestin | |
| Molecular Imager PharosFX | Bio-Rad | |
| Li-Cor Odyssey CLx | Li-Cor | |

# Methods and protocols

## Plasmids and protein purification

The genes encoding the RD domains of DTX1 (388–620), DTX2 (390–622), DTX3 (148–347), DTX4 (387–619) and DTX3L (544–740) were transferred from pDONR221 into the pDEST15 vector to generate N-terminal GST-tag proteins, which were expressed in *E. coli* Rosetta (DE3) cells in LB, and cultures were induced with 300 μM IPTG when the optical density at 600 nm (OD600) reached 0.6 to 0.8 and expressed at 18 °C overnight. Harvested cells were resuspended in lysis buffer (1x PBS containing 4 mM 2-mercaptoethanol). For purification, cell suspensions were thawed, supplemented with benzonase (Novagen) and cOmplete EDTA-free protease inhibitor cocktail (Roche) and lysed by end-over-end mixing for 1 h at 4 °C, followed by EmulsiFlex-C5 homogenizer (Avestin). Lysate was cleared by centrifugation for 60 min at $35,000 \times g$ and incubated with GST agarose for 1 h at 4 °C. Resins were washed with lysis buffer, followed by protein elution with 100 mM Tris-HCl (pH 8.0), 200 mM NaCl, 1 mM dithiothreitol (DTT), and 20 mM Glutathione. All proteins were further purified by size exclusion chromatography (SEC) on a Superdex 200 column (GE Healthcare) before snap-freezing in liquid nitrogen and storing at −80 °C.

The genes encoding full length of human JOSD1 was transferred from pDONR221 into pDEST17 vector to generate N-terminal His$_6$-tag protein. JOSD1 protein was expressed in *E. coli* Rosetta (DE3) cells in LB, and cultures were induced with 300 μM IPTG when the OD600 reached 0.6 to 0.8 and expressed at 18 °C overnight. Harvested cells were resuspended in lysis buffer (25 mM Tris-HCl (pH 7.5), 500 mM NaCl, 1 mM DTT). For purification, cell suspensions were thawed, supplemented with benzonase (Novagen) and cOmplete EDTA-free protease inhibitor cocktail (Roche) and lysed by end-over-end mixing for 1 h at 4 °C, followed by EmulsiFlex-C5 homogenizer (Avestin). Lysate was cleared by centrifugation for 60 min at $35,000 \times g$ and incubated with Ni$^{2+}$ resin for 1 h at 4 °C. Resins were washed with lysis buffer containing 30 mM imidazole, followed by protein elution with lysis buffer containing 300 mM imidazole. The proteins were further purified by size exclusion chromatography (SEC) on a Superdex 200 column (GE Healthcare) before snap-freezing in liquid nitrogen and storing at −80 °C.

WT and mutants of the N-terminal His$_6$-RD domains of DTX3L and DTX2 were expressed and purified as previously described (Zhu et al, 2023; Zhu et al, 2022).

OTULIN, OTUD2 and OTUB1 proteins were purified as previously described (Elliott et al, 2014; Mevissen et al, 2013; Michel et al, 2015). Briefly, these proteins were expressed in *E. coli* strain Rosetta2 (DE3). Cells were grown at 37 °C in 2xYT and purified by affinity chromatography using Ni$^{2+}$ resin or GST agarose. Tags were removed by incubation with PreScission protease or 3 C protease. Further purification was performed by anion exchange and/or size exclusion chromatography.

WT and catalytically inactive mutants of SARS-CoV-2 PLpro and USP2 were produced recombinantly before in our laboratory. Full-length DTX3L (DTX3L fl) was expressed and purified as previously described (Zhu et al, 2023; Zhu et al, 2022).

UBE1 (E-304-050), UBCH5A (E2-616-100), and recombinant Ub (U-100H-10M) were purchased from R&D Systems. Exo T nuclease (M0265S) was purchased from NEB.

## Oligonucleotides

Single-stranded (ss) DNA or RNA oligos used in this study were commercially ordered from Sigma-Aldrich. Oligonucleotides were dissolved to 100 μM stock in 20 mM HEPES–KOH (pH 7.6) and 50 mM KCl buffer.

Sequence of oligonucleotides used in this study (5′ → 3′):
5Cy3_E21_DNA_A [Cy3]GTGGCGCGGAGACTTAGAGAA (PAGE)
5Cy3_E21_DNA_T [Cy3]GTGGCGCGGAGACTTAGAGAT (PAGE)
5Cy3_E21_DNA_G [Cy3]GTGGCGCGGAGACTTAGAGAG (PAGE)
5Cy3_E21_DNA_C [Cy3]GTGGCGCGGAGACTTAGAGAC (PAGE)
5Cy3_E21_DNA_A_3′P [Cy3]GTGGCGCGGAGACTTAGAG AA[Phos] (PAGE)
5Cy3_E21_DNA_A_3′P [Cy3]GTGGCGCGGAGACTTAGAGAA [Phos] (HPLC)
5Cy3_E21_RNA_A [Cy3]GUGGCGCGGAGACUUAGAGAA (PAGE)

## Nucleic acids ubiquitylation assay

4 μM His$_6$-DTX3L-RD or GST-DTX1-RD, DTX2-RD, DTX3-RD, DTX4-RD and DTX3L-RD was incubated with 0.5 μM UBE1, 2.5 μM UBCH5A, 10 μM Ub, and 1 μM Cy3-labelled individual nucleic acids in 50 mM HEPES pH 7.5, 50 mM NaCl, 5 mM MgCl$_2$, 1 mM DTT, and 1 mM ATP. After incubation at 37 °C for 1 h, reactions were split into two equal parts. One half was stopped by addition of 4X LDS sample buffer (Life Technologies) and analysed by SDS-PAGE gel, which was first imaged using the Molecular Imager PharosFX system (BioRad) with laser excitation for Cy3 at 532 nm, then stained with Coomassie staining. Another half was stopped by addition of 2x TBE urea sample buffer (8 M urea, 20 μM EDTA pH 8.0, 20 μM Tris-HCl pH 7.5, and bromophenol blue) and loaded on a pre-run 20% denaturing urea PAGE gel. The gels were run at 6 W/gel and followed by Cy3 visualization using the Molecular Imager PharosFX system (BioRad).

For the hydrolysis assay, 50 mM EDTA was used to stop the ubiquitylation reactions and then various hydrolases were added for a further 30-min treatment. All reactions were split into two parts and stopped, analysed as above mentioned.

For Fig. EV1, 4 μM DTX3L-RD or 0.5 μM DTX3L fl was used.

## PAR ubiquitylation assay

PAR was detached with potassium hydroxide from tankyrase auto-ADP-ribosylation reaction with histones and purified using the dihydroxy boronyl column. The resulting bulk polymers were fractionated by anion exchange chromatography and desalted to yield PAR of defined lengths. (Barkauskaite et al, 2013; Tan et al, 2012) For PAR ubiquitylation, 4 μM His$_6$-DTX3L-RD or DTX2-RD was incubated with 0.5 μM UBE1, 2.5 μM UBCH5A, 10 μM Ub spiked with 1 μM fluorescein-Ub (FLR-Ub) (Cambridge Bioscience), and 50 μM PAR-6mer in 50 mM HEPES pH 7.5, 50 mM NaCl, 5 mM MgCl$_2$, 1 mM DTT, and 1 mM ATP. After incubation at 37 °C for 1 h, the reaction products were analysed by SDS-polyacrylamide gel electrophoresis (SDS-PAGE) with Coomassie staining and Bio-Rad Gel Doc XR System to visualize FLR-Ub modified products.

## Autoubiquitylation assay and NAD$^+$-ubiquitylation assay

For Appendix Fig. S9E, reactions were performed in 50 mM Tris-HCl pH 8.0, 50 mM NaCl, 2.5 mM MgCl$_2$, and 2.5 mM ATP. For autoubiquitylation, each reaction contains 0.2 μM E1, 5 μM

UbcH5A, 50 μM Ub, and 1 μM of the indicated DTX3L-RD, DTX3L-RD$^{ins\_AR}$, DTX2-RD, or DTX2-RD$^{del\_AR}$. For NAD$^+$-ubiquitylation, reactions were supplemented with 100 μM NAD$^+$ and 6.7 μM biotin-NAD$^+$. After incubation at 37 °C for 30 min, the reactions were stopped by adding 4X LDS and 100 mM DTT. The samples were analysed by SDS-PAGE and transferred onto nitrocellulose membrane. The membrane was incubated with ubiquitin (PD41) mouse monoclonal antibody (Santa Cruz), followed by incubation with IRDye® 800CW goat anti-mouse (Li-Cor) and NeutrAvidin DyLight™ 680 (Thermo Fisher) secondary antibodies and scanned using Odyssey CLx (Li-Cor).

### Nucleic acids ubiquitylation blocks Exo T's activity

4 μM DTX3L-RD was incubated with 0.5 μM UBE1, 2.5 μM UBCH5A, 10 μM Ub, and 1 μM Cy3-labelled individual nucleic acids in 50 mM HEPES pH 7.5, 50 mM NaCl, 5 mM MgCl$_2$, 1 mM DTT, and 1 mM ATP. After incubation at 37 °C for 1 h, 10 mM EDTA was used to stop the reactions and then 1U Exo T per reaction was added and incubated at 25 °C for 30 min. The samples were resolved using SDS-PAGE gel and Urea gel and visualized as described above.

### HPLC-MS analyses

HPLC-MS analyses were carried out on an Agilent 1260 Infinity HPLC system, coupled with an Agilent 6120 mass spectrometer [electrospray ionization (ESI) + mode]. The multiply charged envelope was deconvoluted using the charge deconvolution tool in Agilent OpenLab CDS ChemStation software to obtain the average [M] value.

### HPLC-MS monitoring of the enzymatic reactions

12 mM of ADP-ribose (Sigma A0752) or AMP (Santa Cruz Biotechnology sc-214504A) or TMP (Sigma T7004) or GMP (Abcam ab146534) or CMP (Sigma C1006) or 2′-deoxy-ATP (MP Biomedicals 0210054925) or 3′-deoxy-ATP (Santa Cruz Biotechnology sc-396713) were incubated with 5 μM DTX3L-RD, 0.5 μM UBE1, 2.5 μM UBCH5A, and 20 μM Ub in 50 mM HEPES (pH 7.5), 50 mM NaCl, 5 mM MgCl$_2$, 0.5 mM DTT, and 2 mM ATP. Post incubation at 37 °C for 2 h, 10 μl reactions were mixed with 2 μl of 1% TFA. Then reactions were subjected to HPLC-MS analysis as previously described (Zhu et al, 2023; Zhu et al, 2022).

## Data availability

Raw LC-MS data can be accessed with the following link: https://zenodo.org/records/12548162.

The source data of this paper are collected in the following database record: biostudies:S-SCDT-10_1038-S44319-024-00235-1.

## Peer review information

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

## Acknowledgements

We thank Zining Zhu for preparing reagents. The work in I.A.'s laboratory is supported by the Wellcome Trust (210634, 223107, and 302632), Biotechnology and Biological Sciences Research Council (BB/R007195/1 and BB/W016613/1), Ovarian Cancer Research Alliance (813369), Oxford University Challenge Seed Fund (USCF 456), and Cancer Research United Kingdom (C35050/A22284). The work in D.A.'s laboratory is supported by the Edward Penley Abraham Research Fund. MJS is supported by the EU [ERC 101078837] and la Ligue contre le Cancer; he is a fellow of Le Studium and the ATIP-Avenir programme. IG-S gratefully acknowledges support from a Cancer Research UK Career Development Fellowship (C62538/A24670), the Oxford University Press John Fell Research Fund (0006091 and 0014766), and the University of Oxford Medical Sciences Division Pump Priming Award (006353).

## Author contributions

**Kang Zhu**: Investigation; Visualization; Methodology; Writing—original draft; Writing—review and editing. **Chatrin Chatrin**: Investigation; Visualization; Methodology; Writing—original draft; Writing—review and editing. **Marcin J Suskiewicz**: Writing—original draft; Writing—review and editing. **Vincent Aucagne**: Investigation; Writing—review and editing. **Benjamin Foster**: Resources. **Benedikt M Kessler**: Resources. **Ian Gibbs-Seymour**: Resources. **Dragana Ahel**: Conceptualization; Supervision; Funding acquisition. **Ivan Ahel**: Conceptualization; Supervision; Funding acquisition; Writing—original draft; Writing—review and editing.

Source data underlying figure panels in this paper may have individual authorship assigned. Where available, figure panel/source data authorship is listed in the following database record: biostudies:S-SCDT-10_1038-S44319-024-00235-1.

## Disclosure and competing interests statement

The authors declare no competing interests.

# Expanded View Figures

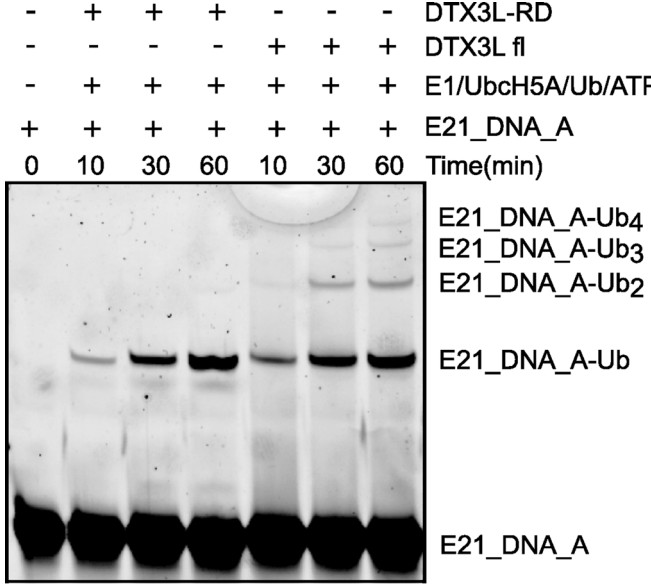

**Figure EV1.  DTX3L catalysed nucleic acids ubiquitylation.**

E21_DNA_A was ubiquitylated by DTX3L-RD and DTX3L fl, at indicated time points. The experiment has been completed in triplicate.

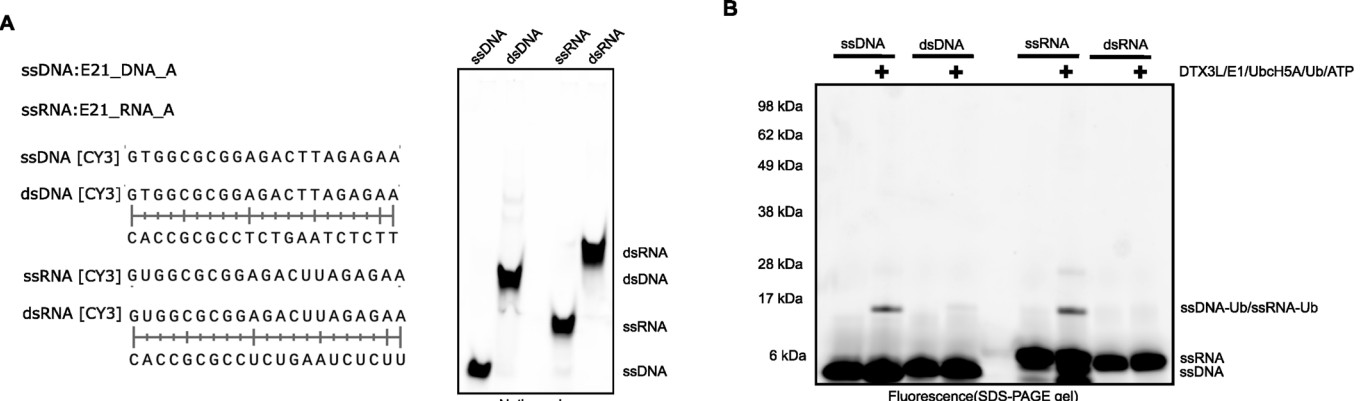

Figure EV2. **DTX3L-RD is not able to ubiquitylate double-stranded nucleic acids.**

(A) Annealed dsDNA and dsRNA were visualised on native gel. (B) dsDNA and dsRNA ubiquitylation assay. The experiment has been completed in triplicate.

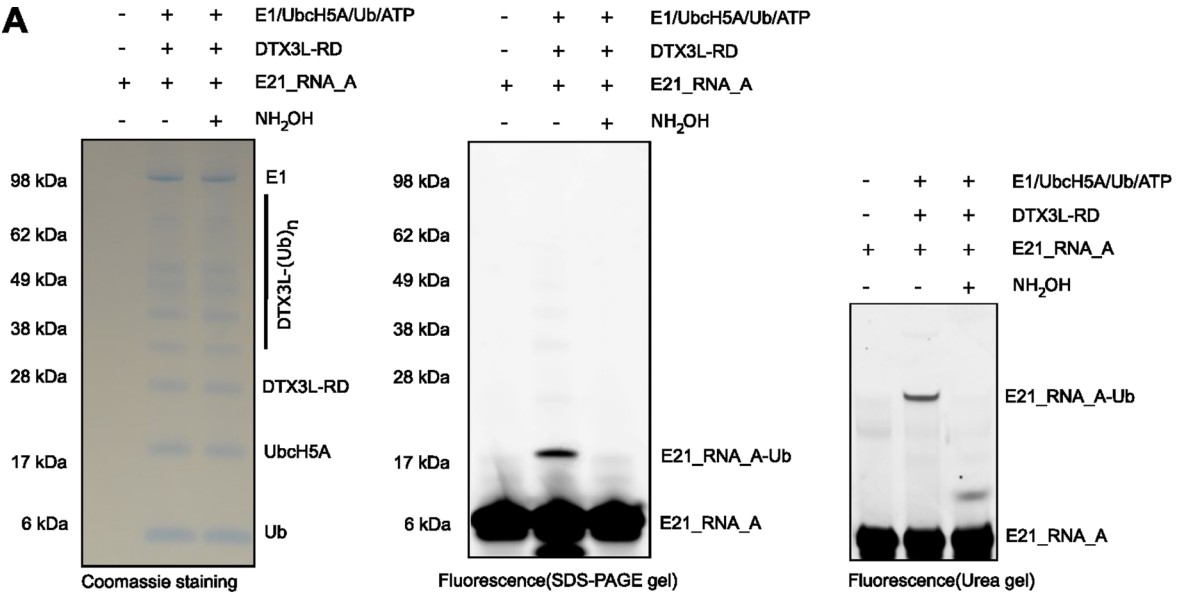

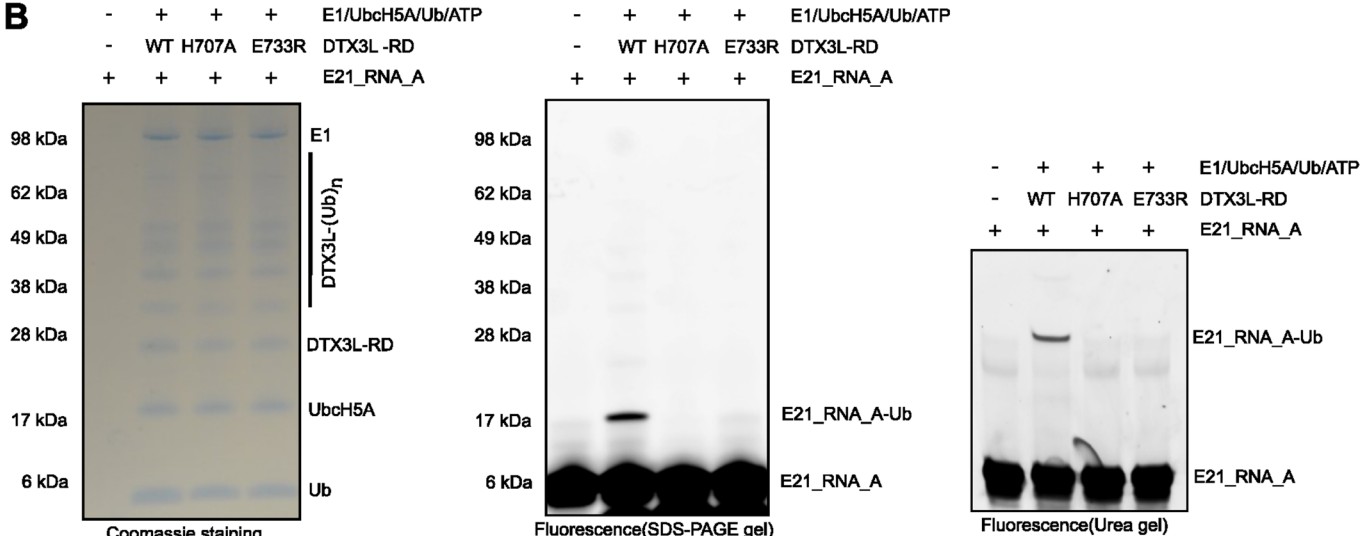

**Figure EV3. DTX3L-RD attaches Ub onto the 3′ hydroxyl group of terminal adenosine in RNA.**

(A) NH$_2$OH reverses DTX3L-RD-catalysed nucleic acids ubiquitylation. NH$_2$OH cleaves the ester bond between the carbonyl group of Gly$^{76}$ of Ub and the 3′ hydroxyl group of the A of E21_RNA_A. (B) DTX3L-RD ADPr ubiquitylation inactive mutants failed to produce upshift bands that correspond to ubiquitylation of RNA, indicating that Ub is attached to 3′ hydroxyl group. Each experiment has been completed in triplicate.

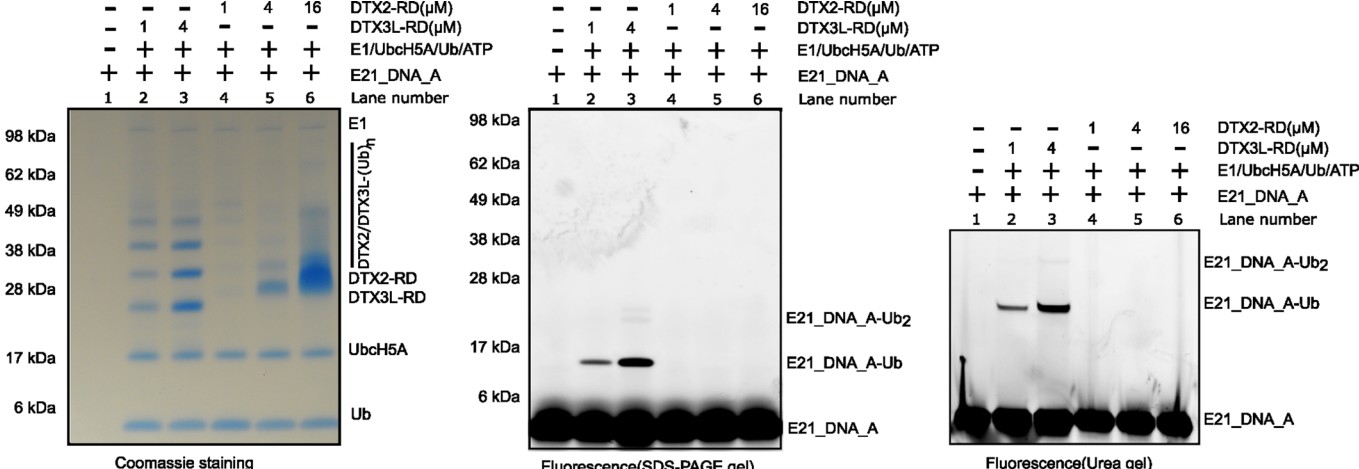

**Figure EV4. DTX2-RD is not able to ubiquitylate nucleic acids.**

E21_DNA_A was incubated with E1, E2 UbcH5A, ATP, Ub and increasing amount of either DTX3L-RD or DTX2-RD, then the reactions were analysed on an SDS-PAGE gel and Urea gel and visualized using the Molecular Imager PharosFX system (BioRad). The experiment has been completed in triplicate.

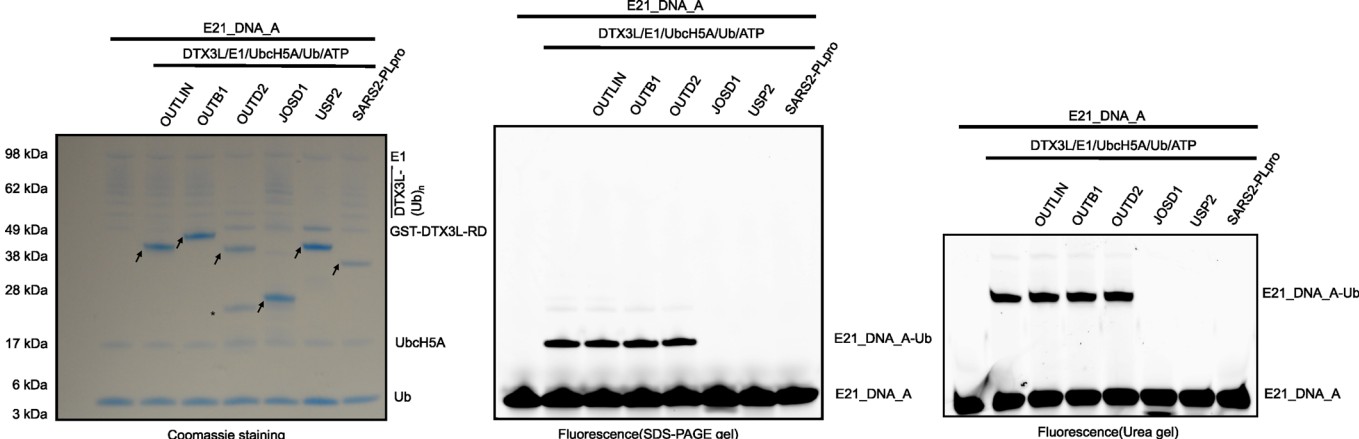

**Figure EV5. DNA ubiquitylation reactions treated with a panel of DUBs.**

Arrows indicate DUBs, asterisk indicates contaminant band.

