## [Peer Review File · EMBO Reports]

Ubiquitylation of nucleic acids by DELTEX ubiquitin E3 ligase DTX3L

Kang Zhu, Chatrin Chatrin, Marcin Suskiewicz, Vincent Aucagne, Benjamin Foster, Benedikt Kessler, Ian Gibbs-Seymour, Dragana Ahel, and Ivan Ahel

Corresponding author(s): Ivan Ahel (ivan.ahel@path.ox.ac.uk), Dragana Ahel (dragana.ahel@path.ox.ac.uk), Kang Zhu (kang.zhu@path.ox.ac.uk)

Review Timeline:

Submission Date:	14th Mar 24
Editorial Decision:	11th Apr 24
Revision Received:	28th Jun 24
Editorial Decision:	13th Aug 24
Revision Received:	16th Aug 24
Accepted:	20th Aug 24

Transaction Report:

Dear Ivan,

Thank you for the submission of your research manuscript to our journal. We have now received the full set of referee reports that is copied below.

As you will see, the referees acknowledge that the findings are interesting, but they also note that the physiological or functional relevance of DNA and RNA ubiquitylation remains unclear at this point. In addition to this general concern, the referees also raise a number of technical concerns. I have discussed the reports and concerns further with the referees and we all agreed that it will not be necessary to test physiological relevance at this stage but that - in the absence of physiological data - the biochemistry would have to be expanded and strengthened along the reviewer comments, in order to publish your manuscript as a focused, biochemical proof-of-principle study. Please clearly discuss the limitations regarding functional significance in the manuscript.

Given the constructive comments and the positive feedback, we would like to invite you to revise your manuscript with the understanding that the referee concerns (as detailed above and in their reports) must be fully addressed and their suggestions taken on board. Please address all referee concerns in a complete point-by-point response. Acceptance of the manuscript will depend on a positive outcome of a second round of review. It is EMBO Reports policy to allow a single round of revision only and acceptance or rejection of the manuscript will therefore depend on the completeness of your responses included in the next, final version of the manuscript.

We realize that it is difficult to revise to a specific deadline. In the interest of protecting the conceptual advance provided by the work, we recommend a revision within 3 months (July 11). Please discuss the revision progress ahead of this time with the editor if you require more time to complete the revisions.

I am also happy to discuss the revision further via e-mail or a video call, if you wish.

*******IMPORTANT NOTE:**

We perform an initial quality control of all revised manuscripts before re-review. Your manuscript will FAIL this control and the handling will be delayed IN CASE the following APPLIES:

- 1) A data availability section providing access to data deposited in public databases is missing. If you have not deposited any data, please add a sentence to the data availability section that explains that.
- 2) Your manuscript contains statistics and error bars based on $n=2$. Please use scatter blots in these cases. No statistics should be calculated if $n=2$.

When submitting your revised manuscript, please carefully review the instructions that follow below. Failure to include requested items will delay the evaluation of your revision.*****

- 1) a .docx formatted version of the manuscript text (including legends for main figures, EV figures and tables). Please make sure that the changes are highlighted to be clearly visible.
- 2) individual production quality figure files as .eps, .tif, .jpg (one file per figure). Please download our Figure Preparation Guidelines (figure preparation pdf) from our Author Guidelines pages <https://www.embopress.org/page/journal/14693178/authorguide> for more info on how to prepare your figures.
- 3) a .docx formatted letter INCLUDING the reviewers' reports and your detailed point-by-point responses to their comments. As part of the EMBO Press transparent editorial process, the point-by-point response is part of the Review Process File (RPF), which will be published alongside your paper.
- 4) a complete author checklist, which you can download from our author guidelines (<<https://www.embopress.org/page/journal/14693178/authorguide>>). Please insert information in the checklist that is also

reflected in the manuscript. The completed author checklist will also be part of the RPF.

5) Please note that all corresponding authors are required to supply an ORCID ID for their name upon submission of a revised manuscript (<<https://orcid.org/>>). Please find instructions on how to link your ORCID ID to your account in our manuscript tracking system in our Author guidelines (<<https://www.embopress.org/page/journal/14693178/authorguide#authorshipguidelines>>)

6) We replaced Supplementary Information with Expanded View (EV) Figures and Tables that are collapsible/expandable online. A maximum of 5 EV Figures can be typeset. EV Figures should be cited as 'Figure EV1, Figure EV2' etc... in the text and their respective legends should be included in the main text after the legends of regular figures.

7) Please note that a Data Availability section at the end of Materials and Methods is now mandatory. In case you have no data that requires deposition in a public database, please state so instead of refereeing to the database. See also < <https://www.embopress.org/page/journal/14693178/authorguide#dataavailability>>. Please note that the Data Availability Section is restricted to new primary data that are part of this study.

Additional information on source data and instruction on how to label the files are available <<https://www.embopress.org/page/journal/14693178/authorguide#sourcedata>>.

10) Figure legends and data quantification:
The following points must be specified in each figure legend:

- the name of the statistical test used to generate error bars and P values,
 - the number (n) of independent experiments (please specify technical or biological replicates) underlying each data point,
 - the nature of the bars and error bars (s.d., s.e.m.)
- If the data are obtained from n {less than or equal to} 5, show the individual data points in addition to the SD or SEM.
- If the data are obtained from n {less than or equal to} 2, use scatter blots showing the individual data points.

See also the guidelines for figure legend preparation:
<https://www.embopress.org/page/journal/14693178/authorguide#figureformat>

11) Our journal encourages inclusion of *data citations in the reference list* to directly cite datasets that were re-used and obtained from public databases. Data citations in the article text are distinct from normal bibliographical citations and should directly link to the database records from which the data can be accessed. In the main text, data citations are formatted as follows: "Data ref: Smith et al, 2001" or "Data ref: NCBI Sequence Read Archive PRJNA342805, 2017". In the Reference list, data citations must be labeled with "[DATASET]". A data reference must provide the database name, accession number/identifiers and a resolvable link to the landing page from which the data can be accessed at the end of the reference. Further instructions are available at <<https://www.embopress.org/page/journal/14693178/authorguide#referencesformat>>.

12) All Materials and Methods need to be described in the main text. We would encourage you to use 'Structured Methods', our new Methods format. According to this format, the Methods section should include a Reagents and Tools Table (listing key

reagents, experimental models, software and relevant equipment and including their sources and relevant identifiers) followed by a Methods and Protocols section in which we encourage the authors to describe their methods using a step-by-step protocol format with bullet points, to facilitate the adoption of the methodologies across labs. More information on how to adhere to this format as well as downloadable templates (.doc or .xls) for the Reagents and Tools Table can be found in our author guidelines: < <https://www.embopress.org/page/journal/14693178/authorguide#manuscriptpreparation>>.

An example of a Method paper with Structured Methods can be found here:
<<https://www.embopress.org/doi/10.15252/msb.20178071>>.

13) As part of the EMBO publication's Transparent Editorial Process, EMBO Reports publishes online a Review Process File to accompany accepted manuscripts. This File will be published in conjunction with your paper and will include the referee reports, your point-by-point response and all pertinent correspondence relating to the manuscript.

Kind regards,

Martina

Referee #1:

EMBOR-2024-59201V1

Although protein ubiquitination has been studied in detail, we still know very little about the significance of the ubiquitin system modifying other biomolecules. More and more of these unconventional reactions have been reported, with various biological consequences. Here, the authors expand on their previous findings about DELTEX family ubiquitin ligases, which are known to ubiquitinate polyADP-ribose (PAR). Zhu et al show that one of the DELTEX family members is capable of ubiquitinating ssDNA or RNA. They show a likely attachment site (the 3'-OH group) and analyze the base preference of the reaction. They also show that another DELTEX member, DTX2, cannot catalyze the reaction, thus indicating that there is some specificity in the system. These results are interesting but preliminary. First, no physiological significance or function is attached to the findings. Second, the intriguing possibility that there is a phylogenetic split in the DELTEX members into DNA-specific and PAR-specific members is discussed but not tested. And finally, the biochemical characterization has not been taken far enough. Upon addressing the comments outlined below, the manuscript could be strengthened and would then become appropriate for publication in EMBO Reports.

Detailed comments:

1. The study is entirely done in vitro with purified proteins, so whether the observed reactions have any physiological significance remains unclear. It is interesting that DNA and RNA can be ubiquitinated and - as mentioned above - there seems to be some specificity in the reaction, but without any evidence for the modification in cells it is difficult to justify a biological significance.

2. The authors show that the RD domain of DTX3L works on DNA, but the RD domain of DTX2 doesn't. This correlates with the presence of a DNA-binding domain in DTX3L and its absence in DTX2, although such domains are absent in the protein fragments used for the assays and therefore not responsible for differentiating between the proteins. It would therefore be good if they could expand their analysis to the other DELTEX members to confirm or disprove this dichotomy in the RD domains and its correlation with the other domains present in those members.

3. Even in the absence of any in cellulo assays, the biochemical characterization should be taken further by answering additional mechanistic questions:

- What is the residual activity on the 3'-phosphorylated oligo due to? Can they abolish it by using a 3'-deoxy-terminus? This is important for RNA as well because in principle the modification could shift to the 3'-OH if the 3'-end is phosphorylated.
- Will dsDNA or dsRNA be modified?
- What are the kinetic parameters towards PAR or ADP-ribose versus DNA or RNA? Does the E3 actually prefer DNA/RNA or is it equally active on both? Experiments using time courses at comparable substrate concentrations would help if genuine rate determinations are not possible.
- Is the base preference due to catalytic rate differences or rather to differences in affinity (i.e. k_{cat} or K_M)?

Referee #3:

The canonical ubiquitination has been considered a posttranslational modification on proteins. However, many non-proteinaceous substrates of ubiquitination have been discovered in the past three years. In this manuscript, the authors identify nucleic acids as novel ubiquitination substrates for the first time using in vitro ubiquitination assays. The experiments are well-designed and performed with high quality. The data presented in the manuscript are sufficient to support their demonstrations. Additionally, they also show that the DTX3L E3 ubiquitin ligase and the USP2 and SARS-CoV-2 PLpro deubiquitinases mediate the reversible ubiquitination of nucleic acids. DTX3L prefers to ubiquitylate 3'-terminal adenosine nucleotides over other nucleotides in nucleic acids. The ubiquitination on the 3'-terminal adenosine protects the nucleic acids from degradation by 3'→5' nucleases such as ExoT, suggesting that ubiquitination on nucleic acids might be important to maintain the stability of DNA and RNA. These findings should be of interest to researchers in DNA damage repair and genomic stability, as well as transcriptomics.

Specific comments:

- 1) The DTX3L-RD exhibits nucleic acid ubiquitination activity, while DTX2-RD does not. The authors speculate that the difference in the RD domain may contribute to this. Given that the other member of the KH domain-containing DELTEX E3s, DTX3, has a very similar RD domain, the authors might consider testing whether DTX3-RD domain exhibits nucleic acid ubiquitination activity.
- 2) In the introduction and discussion, the authors stated the two DELTEX E3 ligases based on the presence of the WWE domain vs. KH domain. At the same time, most of the experiments were carried out using the RING-DTC fragment (without the WWE or KH domains), suggesting the selectivity of DTX3L vs. DTX2 is based on the RING-DTC, not the WWE/KH domains. Maybe it would be helpful to discuss and compare the sequence/structure of the RING-DTC between DTX3L vs DTX2 (or even between DTX3 and DTX3L).

Minor comments:

- 1) In the middle panel of Fig. 1D, the lanes are not aligned with the labels.
- 2) Page 5 (Figure 1D), "Full-length DTX3L (DTX3L fl) appears to be more efficient than DTX3L-RD in catalysing E21_DNA_A ubiquitylation, possibly owing to enhanced substrate recruitment through multiple nucleic acids-binding domain (Zhu et al., 2023)(Figure EV4). However, since the minimum catalytic RING-DTC (RD) fragment is proficient enough and easier to produce, this fragment is used throughout the study." The definition of "RD" should be given in the first sentence/first mention.
 - 1) In EV Fig 9B, "DTXorDTX3L-(Ub) n " should be "DTX2 or DTX3L-(Ub) n ".
 - 3) page 10 (discussion), "... and/or the lack of AR insertion in the DTC domain of ...". The definition of AR seems missing.
 - 4) It might be helpful to comment on whether terminal AMP at dsRNA/DNA could also be modified.

Referee #4:

This is a short and focussed manuscript describing in vitro E3 ligase activity of a RING ligase with a nucleic acid binding domain

ligating ubiquitin to adenosine.

The experiments are for the most part high quality and well-controlled. I do wonder if it's important to have a no E1, and a no E2 control separately, rather than as a single control. My concern for this comes from the fact that one of the products of E1 enzymatic activity is an acyl-adenylate with ubiquitin (Ub.AMP) which may be important to rule out as detectable in a highly sensitive MS reading. Perhaps the authors could use a mutation of the catalytic cysteine in E1 which would be unable to transfer the activated ubiquitin to an E2, and compare the abundance of Ub.AMP with what they see in the presence of the DELTEX member they propose is ubiquitinating adenosine.

Likewise in the DUB experiments, although I appreciate the sharp focus of the manuscript, the text states that DUBs 'including USP2 and SARS... deubiquitinate the adenosine' however, I didn't see any other DUBs tested, it might be worth using a catalytic mutant of USP2 as a control.

There is limited speculation on the potential importance of a reversible modification of adenosine by ubiquitin, other than a suggested 'protective' mechanism. If it is protective, how would the removal be regulated? USP2 is known to be a very efficient DUB, removing all ubiquitination from lysines at least, so it may not be the appropriate DUB to use (or at least include some others). I think it is important to include the identity of the E2 (UbcH5a) in the figures and legends. With ~40 E2s, they are not all identical, and it is important for the reader to be able to see easily that the most promiscuous family member has been used.

The finding is interesting, and further expands the repertoire of chemical ligations of ubiquitin - chemically we already know it is possible to ligate ubiquitin to hydroxy groups, sugars, etc, so their hypothesis is a sensible one. I feel the finding is quite a preliminary one, and warrants either some further controls, or some biological investigation, but the latter would be a lengthy undertaking, and is potentially outside the scope of this report.

In terms of readability, I found the concept of hybrid ADPr-Ub modification unclear - do the authors mean that proteins can be modified with both, at separate sites, or do they mean an ADPr molecule has ubiquitin ligated to it, and the whole thing is then modifying something else? Apologies if I have misunderstood something, but I think the authors may be using a shorthand that is clear to them, but somewhat ambiguous to the reader, this one anyway!

I also found myself distracted by missing articles - eg adjacent to THE RING domain, recruit THE E2-Ub conjugate, THE DTC domain contributes one histidine, accommodate THE ADPr molecule.

We would like to thank the editor and reviewers for their positive evaluation of our manuscript and constructive suggestions, which have helped us improve our manuscript. We have performed additional experiments to address the editor and reviewers' concerns and provided a point-by-point response in blue.

Referee #1:

EMBOR-2024-59201V1

Although protein ubiquitination has been studied in detail, we still know very little about the significance of the ubiquitin system modifying other biomolecules. More and more of these unconventional reactions have been reported, with various biological consequences. Here, the authors expand on their previous findings about DELTEX family ubiquitin ligases, which are known to ubiquitinate polyADP-ribose (PAR). Zhu et al show that one of the DELTEX family members is capable of ubiquitinating ssDNA or RNA. They show a likely attachment site (the 3'-OH group) and analyze the base preference of the reaction. They also show that another DELTEX member, DTX2, cannot catalyze the reaction, thus indicating that there is some specificity in the system.

These results are interesting but preliminary. First, no physiological significance or function is attached to the findings. Second, the intriguing possibility that there is a phylogenetic split in the DELTEX members into DNA-specific and PAR-specific members is discussed but not tested. And finally, the biochemical characterization has not been taken far enough. Upon addressing the comments outlined below, the manuscript could be strengthened and would then become appropriate for publication in EMBO Reports.

We appreciate the reviewer's interest in our results and their thoughtful comments on our manuscript. We have conducted new experiments to address the reviewer's comments regarding the biochemical characterization and tested the nucleic acids ubiquitylation ability of KH domain-containing and WWE domain-containing DELTEX E3s (Figure 4D). Given the novelty of nucleic acids ubiquitylation, we agree with the reviewer that investigating its physiological implications is an important addition to our *in vitro* results, and we will pursue this in our future studies. Since biological function of nucleic acids ubiquitylation represents a long-term effort, we think it is beyond the scope of this short report.

Detailed comments:

1. The study is entirely done *in vitro* with purified proteins, so whether the observed reactions have any physiological significance remains unclear. It is interesting that DNA and RNA can be ubiquitinated and - as mentioned above - there seems to be some specificity in the reaction, but without any evidence for the modification in cells it is difficult to justify a biological significance.

We agree that our *in vitro* results warrant further investigation into the biological function and are currently working on the development of better tools/techniques that will hopefully allow studying this process in cells in the future. As the reviewer suggested, we carried out additional experiments to more thoroughly characterize the specificity of the nucleic acids ubiquitylation reaction *in vitro*. As mentioned in the

next answer, our results demonstrate that the nucleic acids ubiquitylation activity is a feature of only some DELTEX E3s, suggesting that it is a specific process.

2. The authors show that the RD domain of DTX3L works on DNA, but the RD domain of DTX2 doesn't. This correlates with the presence of a DNA-binding domain in DTX3L and its absence in DTX2, although such domains are absent in the protein fragments used for the assays and therefore not responsible for differentiating between the proteins. It would therefore be good if they could expand their analysis to the other DELTEX members to confirm or disprove this dichotomy in the RD domains and its correlation with the other domains present in those members.

We thank the reviewer for the insightful suggestion. We have now tested all the RD domains of DELTEX family members, and the results clearly show that the RD domains of DTX3 and DTX3L, but not of other DELTEX proteins, are able to ubiquitylate nucleic acids ending with 3' adenosine nucleotide (3' A), indicating that the RD catalytic fragment alone is sufficient to dictate nucleic acids ubiquitylation (Figure 4D). Interestingly, this ability of DTX3 and DTX3L correlates with the presence of KH domain(s) at their N-termini. In contrast, the RD domains of DTX1, DTX2, DTX4, which harbor PAR-binding WWE domains on their N-terminus, are unable to ubiquitylate nucleic acids. This data supports our proposal that KH domain-containing DELTEX E3s have nucleic acids ubiquitylation ability, which is inherent in the RD domains.

Sequence alignment and structural analysis suggest that the alanine-arginine (AR) insertion in the DTC domains and the loop in RING domains possibly determine the nucleic acids ubiquitylation ability of DELTEX family members (Appendix Figure S9).

3. Even in the absence of any in cellulo assays, the biochemical characterization should be taken further by answering additional mechanistic questions:

- What is the residual activity on the 3'-phosphorylated oligo due to? Can they abolish it by using a 3'-deoxy-terminus? This is important for RNA as well because in principle the modification could shift to the 3'-OH if the 3'-end is phosphorylated.

We thank the reviewer for pointing this out. As we mentioned in the manuscript, we assume that the residual activity on the 3'-phosphorylated oligo (which, on closer inspection, manifests as two distinct bands that could correspond to two different ubiquitylated species) is mainly due to incomplete phosphorylation during synthesis. To test this possibility, we ordered new E21_DNA_A-3P oligonucleotides subjected to HPLC purification (which gives better purity than the previously used PAGE), and compared ubiquitylation of the PAGE-purified and HPLC-purified E21_DNA_A-3P (Figure R1, left panel). When comparing the reactions with PAGE- or HPLC-purified E21_DNA_A-3P with each other, we observed that the upper band is greatly diminished in the reaction performed with the HPLC-purified substrate, indicating that it might represent the ubiquitylation of contaminating nonphosphorylated oligo (E21_DNA_A-Ub), most likely on 3' OH. Since the lower band didn't show any difference between reactions with PAGE- and HPLC-purified substrates, we assume that, in fact, it corresponds to the ubiquitylated phosphorylated oligonucleotide (E21_DNA_A-3P-Ub).

To shed further light on the nature of these two residual products, we repeated the ubiquitylation of the PAGE-purified substrate and treated the products with NH_2OH (Figure R1, right panel). The upper band was removed by NH_2OH treatment, in line with the expected sensitivity of the ester-linked ubiquitylation of contaminating nonphosphorylated oligo. In contrast, the lower band was resistant to NH_2OH treatment, suggesting that the ubiquityl moiety is attached to the phosphorylated oligo neither through a hydroxyl group nor a phosphate group (an ester or a phosphoanhydride bond would be NH_2OH -sensitive). Instead, the lower band could represent low-efficiency ubiquitylation of the phosphorylated oligo on the NH_2 moiety of one of the bases. However, it is important to stress that this reaction is markedly less efficient than the modification of nonphosphorylated oligo on 2' hydroxyl.

Figure R1. Low abundance ubiquitylation of incompletely phosphorylated oligonucleotide.

Left: ubiquitylation of E21 DNA_A-3P purified by PAGE and HPLC, respectively. Right: NH_2OH treatment of DNA A-3P ubiquitylation.

In addition, we inquired several suppliers about the possibility of synthesizing 3'-deoxy-terminus oligonucleotides; unfortunately, this is currently not possible. As an alternative, we performed mass spectrometry analysis using 2'-deoxy ATP and 3'-deoxy ATP (Figure R2, also Appendix Figure S4 and S5). Our results showed that only 2'-deoxy ATP can be ubiquitylated but not the 3'-deoxy ATP. This is in line with our proposal that ubiquitylation of nucleic acids is conjugated to the 3'-OH ribose of the 3' terminal nucleotide.

DTX3L-RD + 2'-dATP

Compound	Relative abundance (%)	Detected average mass	Identified product, its theoretical average mass
A	31	9036.4	Ub-2'dATP, 9037.9 Da
B	21	9052.3	Ub-ATP, 9053.9 Da
C	13	8956.4	-
D	13	8972.6	Ub-ADP, 8973.9 Da
E	12	8563.2	Ub, 8564.7 Da
F	10	8699.2	Ub-DTT, 8701.1 Da

DTX3L-RD + 3'-dATP

Compound	Relative abundance (%)	Detected average mass	Identified product, its theoretical average mass
A	31	8563.5	Ub, 8564.7 Da
B	25	9052.3	Ub-ATP, 9053.9 Da
C	17	8892.6	Ub-AMP, 8893.9 Da
D	17	8972.5	Ub-ADP, 8973.9 Da
E	10	9699.0	Ub-DTT, 8701.1 Da

Figure R2. LC-MS analysis of reactions with 2'-dATP and 3'-dATP

Top: LC-MS analysis of reaction performed with 2'-dATP

Bottom: LC-MS analysis of reaction performed with 3'-dATP

- Will dsDNA or dsRNA be modified?

We thank the reviewer for pointing this out. We tested blunt dsDNA and dsRNA, using ssDNA and ssRNA ending with 3' adenosine as positive controls. Neither dsDNA nor dsRNA were ubiquitylated by DTX3L-RD, while both ssDNA and ssRNA were ubiquitylated (EV Figure 2). This suggests that DTX3L-RD specifically ubiquitylates ssDNA and ssRNA.

- What are the kinetic parameters towards PAR or ADP-ribose versus DNA or RNA? Does the E3 actually prefer DNA/RNA or is it equally active on both? Experiments using time courses at comparable substrate concentrations would help if genuine rate determinations are not possible.

We thank the reviewer for the helpful suggestion. To compare ubiquitylation efficiency on ssDNA and ssRNA by DTX3L, we performed a time course experiment on 1 μ M ssDNA and ssRNA as the reviewer kindly suggested (Figure R3). We did not observe notable difference in the modification efficiency between these two substrates.

Figure R3. Time course ubiquitylation of ssDNA and ssRNA

To figure out DTX3L's preference for ADP-ribose (ADPr) or DNA, we conducted competition assays by adding 1, 5, or 20 molar excess of competitor to the substrate ubiquitylation reactions (adding ADPr into E21_DNA_A ubiquitylation reactions, or adding E21_DNA_A into ADPr ubiquitylation reactions). We observed that 5 molar excess of ADPr nearly abolished E21_DNA_A ubiquitylation (Figure R4). However, even 20 molar excess of E21_DNA_A had limited effect on ADPr ubiquitylation reactions (Figure R5). These data suggest that DTX3L prefers ADPr over DNA as a substrate.

Figure R4. E21_DNA_A ubiquitylation reactions with ADPr added as a competing substrate.

Figure R5. ADPr ubiquitylation reactions with E21_DNA_A added as a competing substrate. Ub is shown in red, poly/mono-ADPr is shown in green.

Is the base preference due to catalytic rate differences or rather to differences in affinity (i.e. k_{cat} or K_M)?

We tried to measure binding affinity of ADP-ribose and AMP for DTX3L-RD, unfortunately we detected no reliable/measurable binding by ITC. However, there are solved crystal structures of the RING-DTC (RD) domain of DTX1 and DTX2 bound to NAD^+ or ADP-ribose, both of which have an adenosine base (Figure 1A), and the RD domains catalyse Ub transfer to ADPr and NAD^+ *in vitro*. Since the catalytic RD domains is efficient in ADP-ribose ubiquitylation, we think this translates to RD domains preference for adenosine as opposed to other mononucleotides.

Figure R6. DTC domain interaction with ADPr (top) or CDPr (bottom)

In the solved crystal structure of DTX2-RD + ADP-ribose (PDB: 6Y3J), ADP-ribose has extensive contact with the residues in the DTC domain (Figure R6, top panel). In particular, the adenine base is stacked between residues H594 and W578. This stacking with histidine and tryptophan might be suboptimal for pyrimidine bases. Additionally, the adenine N1 is stabilised by hydrogen bond with residue T585. This hydrogen bond interaction is most likely lost with pyrimidine-containing bases (T/C), however is retained with purine bases (A/G). This would suggest that the DTC domain probably binds A/G better compared to T/C, and also in line with our observations that DTX3L catalysed DNA ubiquitylation better on A/G, but inefficient on T/C (Figure 3, both LC-MS and gel-based assay). The RD domain has some preference to bind adenine-containing species, which should translate into a difference in K_M .

However, we cannot rule out that there is an additional k_{cat} component to the difference, for example stemming from the fact that non-adenine nucleotides might

be bound not only more weakly, but also in a suboptimal shifted orientation that makes the reaction less efficient.

Referee #3:

The canonical ubiquitination has been considered a posttranslational modification on proteins. However, many non-proteinaceous substrates of ubiquitination have been discovered in the past three years. In this manuscript, the authors identify nucleic acids as novel ubiquitination substrates for the first time using in vitro ubiquitination assays. The experiments are well-designed and performed with high quality. The data presented in the manuscript are sufficient to support their demonstrations. Additionally, they also show that the DTX3L E3 ubiquitin ligase and the USP2 and SARS-CoV-2 PLpro deubiquitinases mediate the reversible ubiquitination of nucleic acids. DTX3L prefers to ubiquitylate 3'-terminal adenosine nucleotides over other nucleotides in nucleic acids. The ubiquitination on the 3'-terminal adenosine protects the nucleic acids from degradation by 3'→5' nucleases such as ExoT, suggesting that ubiquitination on nucleic acids might be important to maintain the stability of DNA and RNA. These findings should be of interest to researchers in DNA damage repair and genomic stability, as well as transcriptomics.

We thank the reviewer for recognizing the significant findings of our study and providing constructive feedback.

Specific comments:

1) The DTX3L-RD exhibits nucleic acid ubiquitination activity, while DTX2-RD does not. The authors speculate that the difference in the RD domain may contribute to this. Given that the other member of the KH domain-containing DELTEX E3s, DTX3, has a very similar RD domain, the authors might consider testing whether DTX3-RD domain exhibits nucleic acid ubiquitination activity.

We thank the reviewer for the constructive suggestion. We have now tested the RING-DTC domains of all DELTEX family members, and the result clearly showed that DTX3-RD and DTX3L-RD could ubiquitylate nucleic acids ending with 3' adenosine nucleotide (3' A)(Figure 4D). According to the sequence alignment and structural analysis (Appendix Figure S9), we speculate that the alanine-arginine (AR) insertion in the DTC domains and the loop in RING domains possibly define the nucleic acids ubiquitylation ability.

The nucleic acids ubiquitylation ability of DTX3 and DTX3L's RD domains also correlates with the presence of KH domain(s) on their N-terminus. In contrast, the RD domains of DTX1, DTX2, DTX4, which harbor WWE domains on their N-terminus, are unable to ubiquitylate nucleic acids. This new data supports our proposal that KH domain containing DELTEX E3s have the nucleic acids ubiquitylation ability, which is inherent in the RING-DTC domains.

2) In the introduction and discussion, the authors stated the two DELTEX E3 ligases based on the presence of the WWE domain vs. KH domain. At the same time, most of the experiments were carried out using the RING-DTC fragment (without the

WWE or KH domains), suggesting the selectivity of DTX3L vs. DTX2 is based on the RING-DTC, not the WWE/KH domains. Maybe it would be helpful to discuss and compare the sequence/structure of the RING-DTC between DTX3L vs DTX2 (or even between DTX3 and DTX3L).

We thank the reviewer for this constructive suggestion. As mentioned above, we have now tested a full panel of RD fragments of DELTEX proteins, confirming that nucleic acids ubiquitylation can only be catalyzed by KH-containing family members. This difference in activity correlates not only with the presence of the KH domains, but also with structural features in the catalytic RD fragments themselves, which makes sense, because the activities are different already at the level of RD fragments. By analysing the amino acids sequences of the RD fragments, we noticed a slightly longer RING domain and an alanine-arginine (AR) insertion in the DTC domain of WWE domain-containing DELTEX E3s, compared to the KH domain-containing DELTEX E3s (Appendix Figure S9A-C). We suspected that the AR loop in DTX1, DTX2 and DTX4 might account for the loss of nucleic acids ubiquitylation activity (Appendix Figure S9A, C). To test this, we inserted AR to DTX3L-RD (DTX3L-RD^{ins_AR}) and deleted AR of DTX2-RD (DTX2-RD^{del_AR}) and performed the ubiquitylation reactions with DTX3L-RD^{ins_AR} and DTX2-RD^{del_AR} mutants. Gratifyingly, DTX3L-RD^{ins_AR} lost the nucleic acids ubiquitylation activity. However, DTX2-RD^{del_AR} did not acquire the capability for nucleic acids ubiquitylation (Appendix Figure S9D), suggesting more changes are required to change specificity to nucleic acids. These mutants were both active in auto-ubiquitylation but were impaired in ADP-ribose ubiquitylation (Appendix Figure S9E). These results suggest that the presence of the AR loop might represent one structural difference responsible for the inability of some DELTEX E3s to ubiquitylate nucleic acids. However, there seem to be other important differences within the DTC and/or RING domains of the two DELTEX subclasses, which appear to have slightly diverged in their structure to adapt to their partially different functions, including the presence or absence of the nucleic acids ubiquitylation activity.

Minor comments:

1) In the middle panel of Fig. 1D, the lanes are not aligned with the labels.

We have aligned the lanes with the labels.

2) Page 5 (Figure 1D), "Full-length DTX3L (DTX3L fl) appears to be more efficient than DTX3L-RD in catalysing E21_DNA_A ubiquitylation, possibly owing to enhanced substrate recruitment through multiple nucleic acids-binding domain (Zhu et al., 2023)(Figure EV4). However, since the minimum catalytic RING-DTC (RD) fragment is proficient enough and easier to produce, this fragment is used throughout the study." The definition of "RD" should be given in the first sentence/first mention.

We have now defined 'RD' in the first sentence.

1) In EV Fig 9B, "DTXorDTX3L-(Ub)n" should be "DTX2 or DTX3L-(Ub)n".

We have corrected this figure, which is now Expanded View Figure 4.

3) page 10 (discussion), "... and/or the lack of AR insertion in the DTC domain of ...". The definition of AR seems missing.

We have given the definition of 'AR' in the first sentence.

4) It might be helpful to comment on whether terminal AMP at dsRNA/DNA could also be modified.

We showed that dsRNA and dsDNA ending with 3' adenosine cannot be ubiquitylated by DTX3L-RD (EV Figure 2).

Referee #4:

This is a short and focussed manuscript describing in vitro E3 ligase activity of a RING ligase with a nucleic acid binding domain ligating ubiquitin to adenosine.

The experiments are for the most part high quality and well-controlled. I do wonder if it's important to have a no E1, and a no E2 control separately, rather than as a single control. My concern for this comes from the fact that one of the products of E1 enzymatic activity is an acyl-adenylate with ubiquitin (Ub.AMP) which may be important to rule out as detectable in a highly sensitive MS reading. Perhaps the authors could use a mutation of the catalytic cysteine in E1 which would be unable to transfer the activated ubiquitin to an E2, and compare the abundance of Ub.AMP with what they see in the presence of the DELTEX member they propose is ubiquitinating adenosine.

We thank the reviewer for the positive comments and providing valuable suggestions.

We have now prepared reactions with E1 C632A mutant (Figure R7) and separate controls (without E1, without E2, without E3, and without Ub) (Figure R8). We observe a complete conversion of Ub into Ub-AMP only in the reaction containing DTX3L-RD + AMP. In the no AMP control reaction, a small amount of Ub-AMP was detected, as well as Ub-DTT and Ub-glycerol byproducts. In reactions with E1 C632A mutant, which still generates the activated Ub.AMP acyl adenylate conjugate but is unable to transfer this to E2, only negligible amount of Ub-AMP is detected. In control reactions without E1, without E2, and without Ub, no Ub-AMP is detected, and only a negligible amount of Ub-AMP was observed with no E3 control. These data suggest that the Ub-AMP product detected in LC-MS reading is mostly dependent on DTX3L-RD, and not due to E1-generated Ub.AMP conjugate.

Figure R7. LC-MS of E1 mutant reactions

Figure R8. LC-MS control reactions

Likewise in the DUB experiments, although I appreciate the sharp focus of the manuscript, the text states that DUBs 'including USP2 and SARS... deubiquitinate the adenosine' however, I didn't see any other DUBs tested, it might be worth using a catalytic mutant of USP2 as a control.

There is limited speculation on the potential importance of a reversible modification of adenosine by ubiquitin, other than a suggested 'protective' mechanism. If it is protective, how would the removal be regulated? USP2 is known to be a very efficient DUB, removing all ubiquitination from lysines at least, so it may not be the appropriate DUB to use (or at least include some others). I think it is important to include the identity of the E2 (UbcH5a) in the figures and legends. With ~40 E2s, they are not all identical, and it is important for the reader to be able to see easily that the most promiscuous family member has been used.

We thank the reviewer for the insightful suggestion. We now included the catalytic mutants of both USP2 and SARS2-PLpro and compared them to the corresponding wild-type proteins. The mutants failed to cleave ubiquitylated DNA/RNA, indicating that the DNA/RNA ubiquitylation reversal is dependent on DUB catalytic activities (Figure 5B, C).

Additionally, we tested more DUBs, including OTULIN, OTUB1, OTUD2 that show isopeptidase activity and JOSD1 which possesses esterase activity (EV Figure 5). In line with the fact that the linkage between DNA/RNA's 3' terminal adenosine base and ubiquitin is ester bond, we found that JOSD1 reversed DNA ubiquitylation. Meanwhile, OTULIN, OTUB1 and OTUD2 that are reported to specifically cleave one or more types of Ub chains, failed to cleave the ubiquitylated DNA. This data suggests that the reversal of DTX3L-mediated nucleic acids ubiquitylation is likely to be regulated by DUBs with esterase activity.

We have now included the identity of E2 (UbcH5A) in the figure legends.

The finding is interesting, and further expands the repertoire of chemical ligations of ubiquitin - chemically we already know it is possible to ligate ubiquitin to hydroxy groups, sugars, etc, so their hypothesis is a sensible one. I feel the finding is quite a preliminary one, and warrants either some further controls, or some biological investigation, but the latter would be a lengthy undertaking, and is potentially outside the scope of this report.

We appreciate the reviewer recognizing our results interesting and agree that biological investigation is needed in the future, which will take much longer time and is out of the scope of the key results in this study. However, to provide more evidence and solidify our results, following reviewers' insightful suggestions, we performed new experiments with proper controls as shown in the point-by-point response.

In terms of readability, I found the concept of hybrid ADPr-Ub modification unclear - do the authors mean that proteins can be modified with both, at separate sites, or do they mean an ADPr molecule has ubiquitin ligated to it, and the whole thing is then modifying something else? Apologies if I have misunderstood something, but I think

the authors may be using a shorthand that is clear to them, but somewhat ambiguous to the reader, this one anyway!

We apologise for the confusion over our description of hybrid ADPr-Ub. The hybrid ADPr-Ub here refers to DELTEX E3s-mediated attachment of Ub to the ADPr moiety of ADP-ribosylated proteins or nucleic acids by PARPs, thus forming ADPr-Ub modification on protein or nucleic acids substrates. To avoid confusion, we rephrased it in the revised manuscript.

I also found myself distracted by missing articles - eg adjacent to THE RING domain, recruit THE E2-Ub conjugate, THE DTC domain contributes one histidine, accommodate THE ADPr molecule.

We have now added the missing articles into the sentences.

Dear Ivan,

Thank you for the submission of your revised manuscript to EMBO reports. We have received the full set of referee reports that is copied below. As you will see, all referees are very positive about the study and request only minor textual changes (referee 1). Please apologize the delay in the further handling of your manuscript, but I have just now returned to the office.

Browsing through the manuscript myself, I noticed a few editorial things that we need before we can proceed with the official acceptance of your study.

- Your manuscript will be published as short report. The revised manuscript should therefore not exceed 27,000 characters (including spaces but excluding materials & methods and references). The results and discussion sections must further be combined, which will help to shorten the manuscript text by eliminating some redundancy that is inevitable when discussing the same experiments twice.
 - Please provide up to 5 keywords.
 - Please update the 'Conflict of interest' paragraph to our new 'Disclosure and competing interests statement'. For more information see <https://www.embopress.org/page/journal/14693178/authorguide#conflictsofinterest>
 - Regarding the Author Contributions, we now use CRediT to specify the contributions of each author in the journal submission system. Therefore, please remove the Author Contributions from the manuscript file and make sure that the author contributions in our online manuscript tracking system are correct and up-to-date. The information you specified in the system will be automatically retrieved and typeset into the article. You can enter additional information in the free text box provided, if you wish.
 - The ORCID ID is still missing for co-corresponding author Dr. Dragana Ahel. Please find instructions on how to link your ORCID ID to your account in our manuscript tracking system in our Author guidelines (<<https://www.embopress.org/page/journal/14693178/authorguide#authorshipguidelines>)
 - The information on funding needs to be part of the Acknowledgments.
 - Please add headers to the figure legends: "Figure legends" and "Expanded View Figure Legends".
 - We now require that all articles published in EMBO Press contain Structure Methods. In order to comply with this format, we require a Reagents and Tools table listing key reagents, experimental models, software and relevant equipment and including their sources and relevant identifiers) followed by the Methods section. The aim is to facilitate adoption of the methodologies across labs. More information on how to adhere to this format as well as a downloadable template (.docx) for the Reagents and Tools Table can be found in our author guidelines: <https://www.embopress.org/page/journal/14693178/authorguide#structuredmethods>.
- An example of a Method paper with Structured Methods can be found here: <https://www.embopress.org/doi/10.15252/msb.20178071>.
- Figure Legends: Please note that the asterisk and arrows are not defined in the legend of figure EV 5. This needs to be rectified.
 - You mention in the Author Checklist that you specified information on antibodies in the Figures (Materials - Antibodies). If you used antibodies in your study, please include their description in the Methods.
 - The synopsis image will be published with a width of 550 pixels. At this small format, the text in your image is not well legible. Please reformat the image and have a look at whether it looks good if you scale it down to 550 pixels width (the height is flexible, 400-600 pixels). Please upload the downscaled version.
 - I have slightly modified the summary text you sent. Please review the attached document.
 - I have also introduced some small modifications in the Abstract. Please review my suggestion that is pasted at the end of this e-mail
 - On a different note, I would like to alert you that EMBO Press offers a new format for a video-synopsis of work published with us, which essentially is a short, author-generated film explaining the core findings in hand drawings, and, as we believe, can be very useful to increase visibility of the work. This has proven to offer a nice opportunity for exposure i.p. for the first author(s) of the study. Please see the following link for representative examples and their integration into the article web page:

https://www.embopress.org/video_synopses
<https://www.embopress.org/doi/full/10.15252/emboj.2019103932>

With kind regards,

Martina

Referee #1:

The authors have done a thorough job to address the reviewers' concerns. They have added a series of in vitro experiments to extend the characterization of the type of nucleic acid that is ubiquitylated by DTX3L, and they have compared the catalytic domains of all the members of the DELTEX family. With this, they have significantly solidified their study. Therefore, even in the absence of any data addressing the physiological relevance of the phenomenon, I support publication in EMBO Reports. The authors might want to have one more go at eliminating some language issues, but that can potentially also be done at the editorial stage.

Referee #3:

The authors have done an excellent job addressing all my initial comments and concerns. The revised figures and additional data suggest that only KH domain-containing DELTEX E3s possess nucleic acid ubiquitylation activity. The inability of WWE domain-containing DELTEX E3s to ubiquitylate nucleic acids is partially attributed to the presence of the AR-loop in the RD domain. These revisions have significantly improved the overall quality of the manuscript. I believe the paper is suitable for publication.

Referee #4:

The authors have made a great effort to address my previous concerns, including additional experimental data, and edits to discussion. The paper reads well, and in my opinion is suitable for publication.

Proposed Abstract

The recent discovery of non-proteinaceous ubiquitylation substrates broadened our understanding of this modification beyond conventional protein targets. However, the existence of additional types of substrates remains elusive. Here, we present evidence that nucleic acids can also be directly ubiquitylated via ester bond formation. DTX3L, a member of the DELTEX family E3 ubiquitin ligases, ubiquitylates DNA and RNA in vitro and this activity is shared with DTX3, but not with the other DELTEX family members DTX1, DTX2 and DTX4. DTX3L shows preference for the 3'-terminal adenosine over other nucleotides. In addition, we demonstrate that ubiquitylation of nucleic acids is reversible by DUBs such as USP2, JOSD1 and SARS-CoV-2 PLpro. Overall, our study proposes reversible ubiquitylation of nucleic acids in vitro and discusses its potential functional implications.

The authors have addressed all minor editorial requests.

Dr. Ivan Ahel
University of Oxford
Sir William Dunn School of Pathology
South Parks Road
Oxford OX1 3RE
United Kingdom

Dear Dr. Ahel,

I am very pleased to accept your manuscript for publication in the next available issue of EMBO reports. Thank you for your contribution to our journal.

Kind regards,
